# ADAPTIVE TEST-TIME REASONING VIA REWARD-GUIDED DUAL-PHASE SEARCH

## ABSTRACT

Large Language Models (LLMs) have achieved significant advances in reasoning tasks. A key approach is tree-based search with verifiers, which expand candidate reasoning paths and use reward models to guide pruning and selection. Although effective in improving accuracy, these methods are not optimal in terms of efficiency: they perform simple decomposition on the reasoning process, but ignore the planning-execution nature of tasks such as math reasoning or code generation. This results in inefficient exploration of reasoning process. To address this, we propose a dual-phase test-time scaling framework that explicitly separates reasoning into planning and execution, and performs search over the two phases individually. Specifically, we decompose reasoning trajectories and develop reward models for each phase, enabling the search to explore and prune plans and executions separately. We further introduce a dynamic budget allocation mechanism that adaptively redistributes sampling effort based on reward feedback, allowing early stopping on confident steps and reallocation of computation to more challenging parts of the reasoning process. Experiments on both mathematical reasoning and code generation benchmarks demonstrate that our approach consistently improves accuracy while reducing redundant computation.

## 1 INTRODUCTION

In recent years, Large Language Models (LLMs) have achieved remarkable success in complex reasoning tasks such as mathematical problem solving, code generation, and decision making (Chen et al., 2021; Yao et al., 2023). A common approach to improve LLM reasoning is Chain-of-Thought (CoT) prompting (Wei et al., 2022), which guides the model to generate intermediate reasoning steps in a stepwise manner, often improving performance on arithmetic and symbolic tasks.

To further enhance the quality and accuracy of multi-step reasoning, recent works have explored test-time scaling methods, which perform structured search or sampling over multiple reasoning paths. According to Snell et al. (2024), these methods consider two main directions: (1) **Distribution-Based Sampling**, by refining how candidates are generated, e.g., through iterative revision (Muennighoff et al., 2025; Shinn et al., 2023) or parallel sampling with selection (Snell et al., 2024; Diao et al., 2023); and (2) **Reward-Based Searching**, by using verifiers or reward models to select or guide promising reasoning paths (Snell et al., 2024; Wu et al., 2024). Among the latter, Process Reward Modeling (PRM) (Wu et al., 2024) has shown strong performance by evaluating partial reasoning steps and guiding the search process accordingly. Instead of judging only final outcomes, PRM provides fine-grained supervision at the process level, assigning rewards to intermediate steps and enabling the model to distinguish between useful and unproductive reasoning trajectories when generating the intermediate steps. This step-level feedback allows search algorithms to prune low-quality candidates earlier and concentrate computation on promising directions, leading to more efficient and accurate reasoning. While existing PRM-based test-time scaling methods have significantly improved the reasoning performance of LLMs, several key limitations remain.

First, existing literature on test-time scaling methods scales up the computation based on simple decomposition of the whole reasoning process, and there is limited understanding on how test-time scaling behaves beyond the simple decomposition. In particular, complex tasks such as mathematical problem solving and code generation naturally involve two distinct cognitive phases: planning, which entails high-level strategic formulation (e.g., "define variables and set up an equation"),

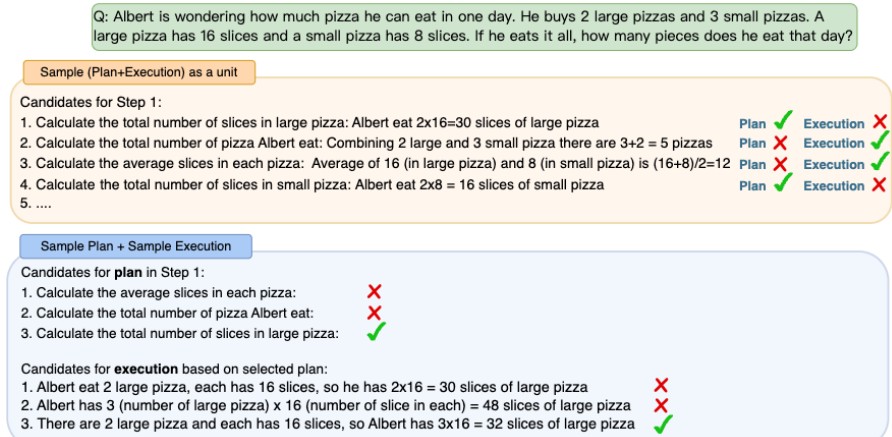

Figure 1: An example of reasoning with plan and execution as a single unit versus searched separately.

and execution, which involves carrying out precise computations or implementations (e.g., arithmetic calculations or code writing) (Zhou et al., 2022; Wang et al., 2024b; Hao et al., 2023; Wang et al., 2023a). Although the benefits of explicitly writing out plans during reasoning have been discussed (Zhou et al., 2022; Wang et al., 2024b; Hao et al., 2023; Wang et al., 2023a), most literature in test-time scaling treats planning and execution as a unified pipeline: the model generates a plan immediately followed by its execution, and both are evaluated together. The consequence is that, if a step has a correct plan but an incorrect execution, the entire candidate is discarded, wasting the useful partial structure. Conversely, if a step is already flawed at the planning stage, the search still wastes budget generating its executions. For example, consider the problem in Figure 1. Since both the plan and the execution may contain errors, sampling them as a unit often requires many trials before obtaining a step in which both are correct simultaneously.

The second limitation of existing tree-based test-time scaling methods is that they mainly adopt a fixed sampling budget per step (e.g., sampling $k$ candidates at every reasoning step), ignoring the varying difficulty across different steps within the same question. This rigid allocation can lead to inefficient computation, especially when simple steps receive excessive attention while more challenging parts remain underexplored. While there are some studies that explore sample-wise budget allocation, i.e., dynamically distributing the overall budget across different questions or different candidate trajectories (Zuo & Zhu, 2025; Lin et al., 2025), these approaches do not address the problem of step-wise allocation within a single reasoning trajectory.

To address these limitations, we propose *DREAM, a Dual-phase REward-guided Adaptive reasoning framework at test tiMe*. Unlike prior methods that treat each plan–execution pair as a single unit, DREAM explicitly conducts search in two stages: it first searches over multiple planning candidates and uses a reward model to select promising subgoals, and then, conditioned on the selected plans, it searches over execution candidates and applies a second reward evaluation to retain the most reliable solutions. For example, for the question in Figure 1, we first sample candidate plans and select the promising ones. Then, conditioned on these plans, we generate multiple execution candidates. This two-stage procedure ensures that poor plans are eliminated early, while promising plans can be paired with different execution attempts until the correct result is found, which ensure that computation is allocated more efficiently across the two phases. In addition to the above, we further incorporates DREAM with a *dynamic budget allocation* mechanism that adaptively adjusts the number of samples at both phases based on real-time reward feedback, enabling early stopping on easy steps and reallocating resources to harder ones.

To verify the effectiveness of the proposed algorithm, we conduct comprehensive evaluations across two domains: math reasoning and code generation. Experimental results show that our approach not only improves answer accuracy but also enhances test-time efficiency.

## 2 RELATED WORKS

**Test-time Scaling.** Test-time scaling methods improve reasoning quality without parameter updates by expending more computation at inference. According to Snell et al. (2024), two primary mechanisms for scaling test-time include (1) Distribution-Based Sampling and (2) Reward-Based Search-

ing. Methods of (1) include s1 (Muennighoff et al., 2025) and Reflexion (Shinn et al., 2023), which introduces sequential self-revision to iteratively refine candidate solutions, and Best-of-N (Wang et al., 2022), which samples multiple candidate reasoning chains in parallel and aggregates via majority voting (self-consistency).

Methods of (2) work by treating intermediate reasoning states as nodes in a search tree and expand continuations via the base LLM. They include MCTS-based methods, such as RAP (Hao et al., 2023), LiteSearch (Wang et al., 2024a), rStar (Qi et al., 2024) and rStar-Math (Guan et al., 2025), which apply Monte Carlo Tree Search to explore reasoning paths, and verifier-based methods, which rely on outcome-level judges (Cobbe et al., 2021; Snell et al., 2024) or process-supervised reward models (PRMs) (Lightman et al., 2023; Wu et al., 2024; Hooper et al., 2025) to score and prune candidates. Moreover, Setlur et al. (2025) indicate that verifier-based methods combined with with search-based strategy are provably better than verifier-free approaches. While effective, most current approaches (even those that adopt a plan-execution format) still treat planning and execution as a single unified process, without performing separate search or adaptive budget allocation across the two phases. Moreover, we would like to highlight that, although a variety of test-time methods have been proposed, in our experiments we mainly consider reward-model-based methods as baselines to ensure a fair comparison, as reward models provide additional information beyond the base LLM.

**Code Generation with LLMs.** Recent work has explored diverse strategies (including test-time scaling) to enhance LLMs for code generation. For example, S*(Li et al., 2025) employs parallel sampling with sequential scaling and adaptive input synthesis to improve code generation. Yu et al. (2025) introduce Z1, which trains the LLM on both short and long reasoning trajectories and leverages a shifted thinking window to enable the model to adaptively control the length of its 'thinking' process according to problem complexity. In addition, PlanSearch (Wang et al., 2024b) boosts code generation by exploring diverse natural-language plans before translating them into code. In addition, tree-structured or agent-based searching frameworks like CodeTree (Li et al., 2024), Tree-of-Code (Ni et al., 2024) and Funcoder Chen et al. (2024) design stepwise generation or refinement algorithms for code generation, where candidate programs are expanded or revised through structured search, demonstrating the benefits of structured exploration.

Although many of the above code generation methods also introduce a planning stage before the generation of code, they typically focus only on improving or selecting a good plan to guide execution rather than performing separate search processes for planning and execution. In contrast, our work performs dual-phase scaling and selection, and assigns budget across phases, which has the potential of having a more effective use of computation.

## 3 PROPOSED METHOD

In this section, we present the general idea and design details of our proposed method. In Section 3.1, we introduce the hierarchical structure of reasoning steps and the decomposition method. In Section 3.2, we introduce the dual-phase search over planning and execution, including both the motivation and detailed implementation. We utilize math reasoning task and code generation task as examples to implement the main idea of the dual-phase search, and introduce the main algorithms for both tasks. In Section 3.3, we describe how to develop the reward models, including both the construction of training data and the design of the reward function.

### 3.1 HIERARCHICAL DECOMPOSITION OF REASONING STEPS

While chain-of-thought prompting and test-time scaling have been shown to significantly improve reasoning performance, existing approaches generally treat reasoning as a flat token-generation process, overlooking the inherent structural characteristics of reasoning.

To address this limitation, following recent latent-variable formulations of reasoning (He et al., 2024; Xie et al., 2021; Tutunov et al., 2023), we consider LLM reasoning considered as a hierarchical process which interleaves two structured reasoning roles: (i) global structural decisions (planning) and (ii) local step derivation (execution). This is similar to the hierarchical structure with the hidden/observed states in a Hidden Markov Model. Our "planning" phase represents the latent structural pattern, and the "execution" phase represents the observable realization of that structure. Specifically, the following are the formal definition grounded in their functional roles in generation:

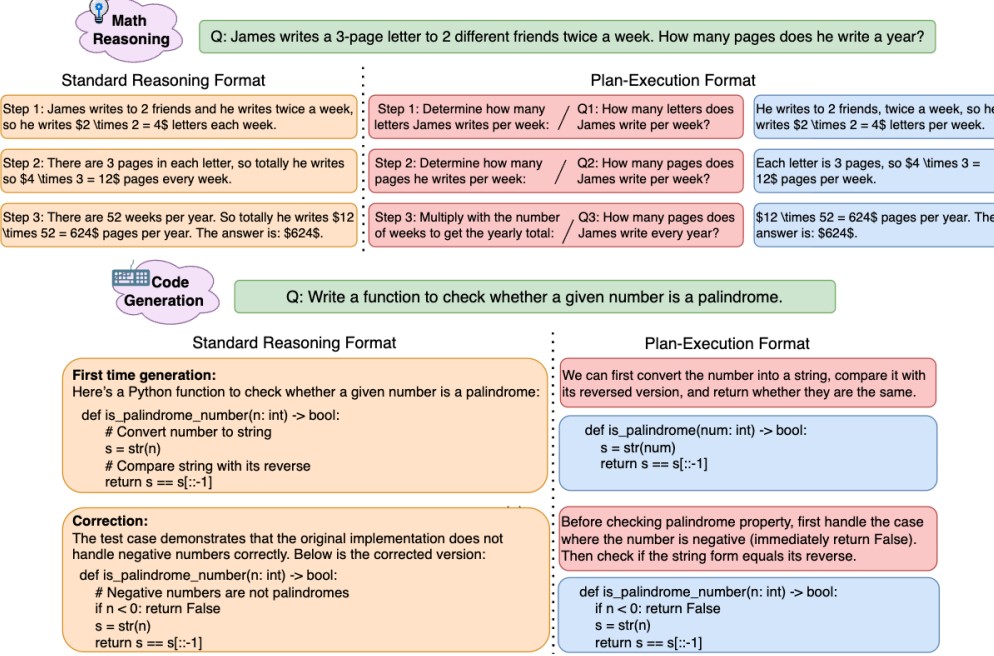

Figure 2: Plan–execution Format for Reasoning in Math and Code Tasks.(typo fixed)

**Definition 3.1 (Planning Step)** *At reasoning step $i$, the planning step produces a structural decision $z_i$ in the form of a token-level description (e.g., subgoal, decomposition strategy, or high-level direction). Formally, it is generated from the model via $z_i \in p_\theta(\cdot|h_i)$, where $p_\theta$ denotes the LLM and $h_i$ is the dialogue history up to step $i$. The sequence $\{z_i\}$ forms a **hierarchical structure** of reasoning intent that organizes and guides the global trajectory of the reasoning process.*

**Definition 3.2 (Execution Step)** *Given the planning $z_i$, the execution step generates the corresponding concrete derivation $x_i$. Formally, it follows $x_i \in p_\theta(\cdot|h_i, z_i)$, where the generation is conditioned on both the prior context and the chosen structural decision. Execution operates at a lower hierarchical level, translating the structural directives in $z_i$ into an explicit local derivations.*

Although reasoning inherently includes structural (planning) and derivational (execution) components as we defined above, current LLMs do not explicitly separate them and these behaviors are often expressed in a mixed and entangled manner within regular generation. In order to make this structure explicit and observable, we utilize few-shot prompts to induce the model to generate reasoning in a separated plan–execution format. As shown in Figure 2, the prompt expresses the *plan* as a sub-question or subgoal, and the *execution* as a direct response that completes that subgoal through concrete derivations. This prompting strategy allows us to disentangle the two reasoning roles during inference, even though the underlying model was not trained with this structure. The whole few-shot prompt can be found in Appendix E.

### 3.2 DUAL-PHASE SEARCH OVER PLANNING AND EXECUTION

Building on the hierarchical decomposition established above, in this subsection we first discuss the empirical observations that reveal distinct behaviors of planning and execution, which motivate the need for a dual-phase search strategy. We then introduce the dual-phase search algorithm in detail. Since the detailed implementation of test-time scaling varies among different type of tasks, following other literature in test-time scaling, e.g., (Hao et al., 2023; Li et al., 2024), we use two representative tasks, math reasoning and code generation, to present our algorithm.

**Motivation for the Dual-phase Search.** The motivation for dual-phase search stems from our observation that planning and execution exhibit fundamentally different behaviors and different error patterns. In short, planning steps are subjected to higher uncertainty, but their errors are usually less fatal. On the other hand, execution is simpler, but the calculation error in the math problem can be carried over in the later steps, being more harmful to the reasoning process. To justify the above, we provide the following two experiment results. **First**, we use the perplexity as a metric to evaluate

Table 1: Distribution of Perplexity and Reward

| | | mean | median | std | min | max |
|---|---|---|---|---|---|---|
| **perplexity** | plan | 5.4646 | 3.8493 | 6.456 | 1.0679 | 88.4054 |
| | exec | 1.2614 | 1.2297 | 0.1831 | 1.0024 | 2.3179 |
| **reward** | plan | 0.804 | 0.957 | 0.274 | 0.008 | 1.000 |
| | exec | 0.768 | 0.957 | 0.319 | 0.002 | 1.000 |

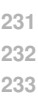

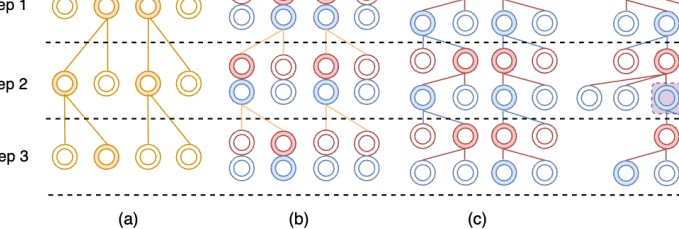

Figure 3: Workflow of Standard Beam Search and Dual-Phase Search (with budget allocation).

the confidence of the LLM towards the planning and the execution step. The results are summarized in Table 1. From the table, it is clear that the perplexity of the planning and execution are different, demonstrating that the LLM is in general less confident in the planning step. **Second**, we also check the distribution of the reward value of the planning and execution steps. The results are also in Table 1. Based on the numbers, the reward for the execution steps are indeed lower than the planning steps. Our conjecture is that, if the planning makes a mistake, the later planning steps can still search and possibly reach the correct result. However, for the execution steps, once a step gives a wrong number, it is very likely that the later steps will use this number for further calculation.

Given the above two differences on the planning and execution steps, we hypothesize that separating them during the inference time and dynamically assigning a computation budget could enhance the efficiency of test-time scaling. In particular, searching over planning steps enables the model to quickly identify promising high-level directions, thereby improving the efficiency of exploration. Meanwhile, searching over execution steps helps the model avoid committing to incorrect derivations and reducing error propagation.

**Dual-phase Search for Math Reasoning.** Our dual-phase search builds on the standard beam search framework, whose workflow is shown in Figure 3 (a). In standard beam search, we utilize the standard reasoning format, where planning and execution are not explicitly expressed, and use the reasoning model to sample a fixed number of candidates. Reward models are applied to score all candidates, and the top-ranked ones are retained for expansion in the next step. Motivated by prior work highlighting the benefits of explicitly separating planning and execution (Zhou et al., 2022; Wang et al., 2024b; 2023a), we further extend standard beam search into a variant that outputs plan–execution pairs in a single step, as illustrated in Figure 3(b).

However, as mentioned in Section 1, the design of treating planning and execution as a single unit is inefficient, since it fails to allocate computation appropriately across the two phases. To overcome this drawback, our dual-phase search (Figure 3(c)) explicitly separates each step into a planning phase (red nodes) and an execution phase (blue nodes). In the planning phase, $N_1$ candidate subgoals are sampled and scored by a planning reward model ($PRM_{plan}$), where the top $n_1$ candidates are selected. In the execution phase, $N_2$ continuations are generated conditional on the chosen plans and scored by an execution reward model ($PRM_{exec}$). Then the top $n_2$ candidates will be selected for the expansion of next step. This separation ensures that the weaker plans can be prune early while promising plans can be given multiple execution attempts, which reduces the risk of discarding good strategies due to execution errors.

In addition, we further extend our method to a *budget-adjusted dual-phase search* that incorporates adaptive allocation. We design this mechanism based on the observation that reasoning difficulty exhibits substantial variance: not only across different problems within a dataset, but also across different steps within the same problem (see examples in Appendix A.2). As a result, allocating

the same sampling budget to every step of every example is inefficient: easy steps waste resources, while difficult steps remain underexplored. To address this limitation, we introduce an adaptive allocation strategy that stops early when confident candidates are already found and reallocates additional computation to more challenging steps, improving the overall accuracy–efficiency trade-off. The specific workflow is shown in Figure 3 (d) and the detailed algorithm can be found in Algorithm 1 in Appendix A.1. At each step, sampling in both the planning and execution phases follows a two-threshold rule. Specifically, as candidates are sampled and scored, if at least $n_1$ (for planning) or $n_2$ (for execution) candidates exceed a specific threshold $\tau_{p1}$ (for planning) or $\tau_{e1}$ (for execution), sampling is terminated early without consuming the full budget. Conversely, if after exhausting the full budget there is no candidate whose reward value higher than a lower threshold $(\tau_{p2}/\tau_{e2})$,[1] we allow at most an additional $m_1$ (planning) or $m_2$ (execution) samples to be generated in order to search harder steps more thoroughly. This mechanism prevents wasted computation on easy steps with confident high-reward candidates, while allocating extra exploration to uncertain or challenging steps. By combining dual-phase scoring with this adaptive budget policy, our method aligns computation with step-level difficulty and improves efficiency over standard beam search (as will be demonstrated in the experiments in Section 4.2). Furthermore, as will be discussed in Appendix C.2, the combination of dual-phase search and dynamic budget allocation has a synergy effect, as the two components mutually reinforce each other by reducing wasted computation and reallocating resources to harder steps.

**Dual-phase Search for Code Generation.** For code generation tasks, we follow the framework of CodeTree (Li et al., 2024) and extend it with our dual-phase search. We adopt this framework because its tree-based structure provides a natural backbone for implementing our dual-phase design. The key difference from math reasoning is that, in code generation task, a visible test set is available for debugging. Instead of treating each step as a partial components of the solution, in CodeTree, each step produces a complete program, and subsequent steps perform iterative debugging based on execution results from failed test cases of earlier solutions.

In the original CodeTree algorithm, both the initial generation and subsequent debugging involve sampling multiple planning candidates per step, which are then attempted sequentially. Whether to expand the current step or backtrack to alternative candidates depends on whether the current node's reward exceeds that of the previous one. The node's reward value is computed by combining two factors: (i) the percentage of passed test cases, and (ii) a score given by an LLM-based critic agent.

We modify the above framework in several ways to implement the dual-phase search. First, for each step, we apply a dedicated reward model to the $N_1$ sampled planning candidates, rank them, and prioritize higher-scoring plans for execution. Second, in the execution phase, we scale generation by producing $N_2$ candidate solutions conditioned on the chosen plan, and select the one with the highest reward. Third, as evidenced by experimental results shown in Appendix C.1, the critic agent provided only marginal benefit. As a result, we remove this component and rely solely on the percentage of passed test cases as the execution reward. Finally, we incorporate a budget-adjustment criterion similar to that in mathematical reasoning: if the reward of a generated candidate exceeds a threshold, we stop further sampling and save the unused budget; if the rewards of all sampled candidates fall below another threshold, additional budget is allocated to generate more candidates. Through these modifications, we extend CodeTree into a dual-phase search framework that conducts separate searches for planning and execution, integrates reward methods for each phase, and allocates computation adaptively based on step-level difficulty.

**Remark.** We note that the general idea of dual-phase search is not limited to math or code reasoning, but can generalize to a wide range of reasoning tasks, as long as the solutions can be expressed in a plan–execution format and the reasoning process can be organized within a tree-based framework.

### 3.3 CONSTRUCTION OF THE REWARD MODELS

**Training data.** To develop the reward models, we follow the general idea of Wang et al. (2023b) to construct datasets that evaluates the quality of both planning and execution at each intermediate reasoning step. We begin by generating complete multi-step reasoning trajectories. For each question in the training set, we sample multiple trajectories at a higher decoding temperature to ensure diversity. Each trajectory is expressed as a sequence of step-wise plan–execution pairs that progressively lead toward the final solution.

---

[1]In practice, for simplicity, we set $\tau_{p1}=\tau_{e1}$ and $\tau_{p2}=\tau_{e2}$

To annotate the steps, we adopt a rollout-based labeling strategy. For each intermediate plan or execution, we generate five independent continuations beginning from that step using the same LLM that produced the trajectory. If at least one of these rollouts leads to a correct final answer, the current step is labeled as positive ("+"); otherwise negative ("–"). This approach assesses the utility of a plan/exeuction based on its downstream impact on solving the problem.

**Reward function:** We implement the reward model by fine-tuning an instruction-tuned LLM. The input to the reward model, denoted as $x$, consists of the original question, all preceding reasoning steps (including both plans and executions), and the current plan or execution to be evaluated. In terms of the output, instead of adding a separate classification head, we follow Dong et al. (2024) to reformulate the prediction as a next-token prediction task: the final position of the input sequence is reserved for a binary label, and the model is trained to output either "+" or "–" at that position. The reward function is then given as:

$$\text{Reward}(x) = \text{softmax}(\ell(x))_+ = \frac{\exp(\ell_+(x))}{\exp(\ell_+(x)) + \exp(\ell_-(x))}$$

where $\ell_+(x)/\ell_-(x)$ are the logits output by the model when predicting the special tokens "+"/"–".

## 4 EXPERIMENTS

### 4.1 SETUP

We evaluate our method in both math-reasoning and code generation tasks. In this subsection, we introduce the experiment setups for both tasks.

**Maths Reasoning.** We evaluate our method on two widely used math reasoning benchmarks: GSM8K (Cobbe et al., 2021) and MATH (Hendrycks et al., 2021). For GSM8K, the full training set of approximately 7.5k problems is used to construct training data for the reward model, while evaluation is performed on the 1.3k test set. For MATH, we leverage the 12.5k training problems to build reward-model training data and conduct evaluation on the standard MATH500 benchmark, a representative subset of 500 problems from the MATH test set. We generate large-scale synthetic trajectories to build up the training dataset: about 400k samples for GSM8K and 400k samples for MATH. The training trajectories and the rollout process for label assignment are produced using LLaMA-3-8B-Instruct (for GSM8K) and Qwen-2.5-3B-Instruct (for MATH). We then combine data from both datasets to train the reward models, fine-tuning Qwen-2.5-32B-Instruct (Yang et al., 2024b). The resulting reward model is applied in experiments on both benchmarks.

We compare our method with three baselines: majority vote, standard beam search, and RE-BASE (Wu et al., 2024), a tree-based search method that does not implement dual-phase search. To ensure a fair comparison, we format all reasoning in the plan–execution style and use the same reward model (trained with the rollout-based annotation strategy) across all methods which involve using reward models. For evaluation, we use three different LLMs on each benchmark: Qwen-2.5-MATH-1.5B-Instruct (Yang et al., 2024a), DeepSeekMath-7B-Instruct Shao et al. (2024), and LLaMA-3-8B-Instruct for GSM8K / LLaMA-3.1-8B-Instruct (Grattafiori & et al., 2024) for MATH. We use different versions of LLaMA for the GSM8K and MATH experiments to ensure that the reasoning models have moderate ability relative to the difficulty of each dataset. This choice allows us to better demonstrate the effectiveness of test-time scaling, since improvements are more evident when the base model is neither too strong nor too weak. More details about the training and inference configurations and the settings of budgets and thresholds can be found in Appendix B.

**Code Generation Reasoning.** For code generation, we conduct experiments on HumanEval (Chen et al., 2021) and MBPP (Austin et al., 2021), including their extended versions (HumanEval+ and MBPP+) (Liu et al., 2023), which include more challenging test cases. The reward model training data is drawn from the MBPP training set (approximately 600 examples), augmented with around 3,000 examples from CodeAlpaca (Chaudhary, 2023). The generation of training trajectories and the labeling process are carried out using a group of Qwen2.5-Coder models ranging in size from 1.5B to 32B. We use models with different capacities to produce a wider range of trajectories (both correct and incorrect), which improves the diversity of supervision for training the reward model. The final reward model is obtained by fine-tuning Qwen-2.5-Coder-7B-Instruct (Hui et al., 2024). We compare our method against the standard CodeTree (Li et al., 2024) and Reflexion Shinn et al. (2023), which prompts the LLM to repeatedly reflect on its previously generated code based on

test case results. The evaluations are conducted with two LLMs including LLaMA-3.1-8B-Instruct, Qwen-2.5-Coder-1.5B-Instruct (Hui et al., 2024).

## 4.2 MAIN RESULTS.

In this subsection, we present the main experimental results to demonstrate the effectiveness of DREAM in achieving a better accuracy-efficiency trade-off in reasoning. The results for math reasoning tasks and code generation tasks are present in Section 4.2.1 and 4.2.2, respectively.

### 4.2.1 MATH REASONING

In this subsection, we present the main results of DREAM in math reasoning tasks. We show the accuracy–tokens frontier of each method in Figure 4. For DREAM, we consider the variants with/without budget allocation, which are labeled "DREAM" and "DREAM(+)", respectively.

Based on the results shown in Figure 4, we have the following observations. **First**, all tree-based search methods consistently achieve a significantly better accuracy–efficiency trade-off than majority vote. For example, on the MATH dataset with LLaMA-3.1-8B-Instruct, the performance gap can reach up to 20%. This confirms the effectiveness of tree-structured search mechanisms in improving the accuracy–efficiency trade-off. In addition, the consistently strong performance across models also highlights the effectiveness of the reward model trained with the rollout-based annotation strategy. **Second**, in general, DREAM outperforms standard beam search which does not explicitly separate planning and execution in searching. For example, on the MATH dataset with Qwen2.5-MATH-1.5B, DREAM continues to outperform beam search, and the advantage becomes more pronounced as the token budget increases. This suggests sampling and selecting planning and execution independently provides a more effective search of reasoning steps and leads to higher-quality candidate trajectories. **Third**, in many of our experiments, DREAM(+) with dynamic budget allocation provides additional gains in accuracy–efficiency trade-off, demonstrating the effectiveness of adaptively allocating computation based on the reward value. For example, on GSM8K with LLaMA-3-8B-Instruct, DREAM(+) consistently achieves about a 2% improvement in accuracy over DREAM at comparable token budgets. While in some cases (often on the MATH dataset), where the problems are more challenging, the improvement over standard dual-phase search is marginal, this can be explained by the fact that the adaptive budget mechanism is more effective when step difficulty is highly variable. When every step in a trajectory is uniformly hard, reallocating budget provides little benefit, and performance is mainly constrained by the inherent capacity of the reasoning model and the reward model. This explanation is supported by the observation that in GSM8K, a large percentage (around 80%) of reasoning steps trigger early stopping, whereas in MATH this percentage is much smaller (around 5%). This suggests that GSM8K contains more diverse step-level difficulty, while MATH problems are more uniformly hard.

**Finally**, we note that in many of our settings, the models used to develop the training data for the reward model are different from the models used in the final evaluation. The only in-distribution setting is LLaMA-3-8B-Instruct with GSM8K, while all other evaluation settings are out of distribution. Nevertheless, in the out-of-distribution settings, DREAM consistently demonstrates strong performance, indicating that our reward model generalizes well across different backbone LLMs rather than overfitting to the one used in training.

### 4.2.2 CODE GENERATION

In this subsection, we present the main results of DREAM combined with CodeTree on code generation benchmarks, comparing with Reflexion and the standard CodeTree method. Figure 5 shows the accuracy–token curve of each approach. Consistent with the math reasoning experiments in Section 4.2.1, we report two variants of DREAM: the version with/without budget allocation (denoted as "DREAM/DREAM(+)").

There are several observations from the experiment. **First,** according to the results in Figure 5, we observe that across all settings and datasets, the performance of CodeTree is significantly improved when combined with dual-phase search using reward models. For example, when the computation budget is around $10^3$ tokens, CodeTree+DREAM achieves an accuracy about 10% higher than CodeTree on both MBPP and HumanEval with Qwen2.5-Coder-1.5B-Instruct. This demonstrates the effectiveness of separated searching of planning and execution and leveraging dedicated reward signals for each phase. In addition, applying budget allocation consistently provides further gains in the accuracy–efficiency trade-off. This suggests that adaptively allocating computation not only

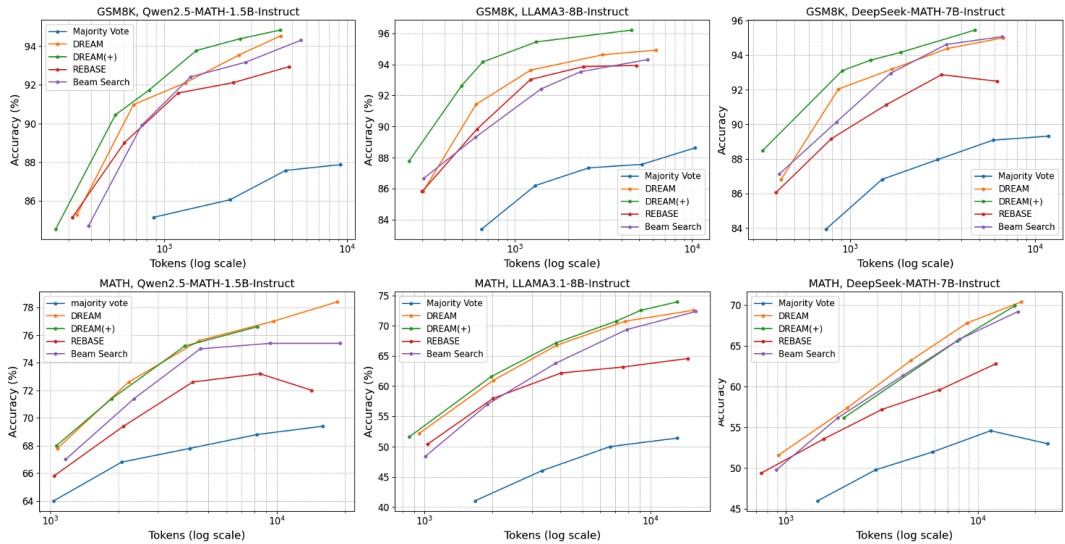

Figure 4: Accuracy vs Tokens (log) on GSM8K/MATH datasets

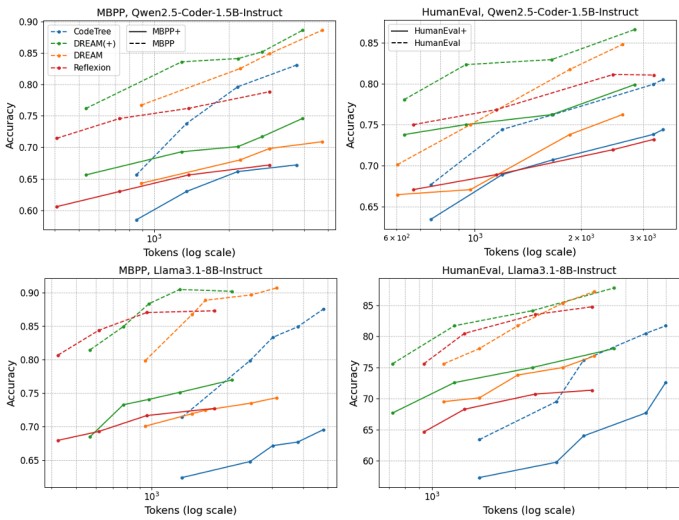

Figure 5: Accuracy vs Tokens (log) on MBPP/HumanEval datasets

improves accuracy but also reduces unnecessary generation cost. **Second,** in some cases, when the token budget is small, Reflexion also achieves a strong accuracy–efficiency trade-off. For instance, with LLaMA-3.1-8B-Instruct, Reflexion performs better when the token budget (log scale) is below $10^3$. However, as the budget increases, the accuracy of Reflexion grows more slowly compared to CodeTree+DREAM, indicating that Reflexion saturates earlier, while dual-phase search continues to benefit from additional computation. **Finally**, it is also worth noting that HumanEval does not appear in the training data of the reward model. The advantage of CodeTree+DREAM in Figure 5 also demonstrates the strong generalization ability of our reward model to unseen datasets.

In short, these findings highlight the advantages of integrating dual-phase search and adaptive budget allocation into tree-based code generation framework.

### 4.3 GENERALIZATION OF THE MATH REWARD MODEL TO UNSEEN DATASETS

In the main results, we demonstrated the transferability of the reward model in the code generation task on unseen datasets, as the HumanEval dataset does not appear in the training data of the reward model. In this section, we further examine the reward model's transferability in the math reasoning domain. Specifically, we consider two out-of-distribution datasets: AMC23 (Math-AI, 2023), which consists of 40 competition-style problems from the American Mathematics Competitions 2023, and the test set of ASDiv (Miao et al., 2021), which contains 301 diverse grade-school math word problems. We evaluate reasoning with DREAM/DREAM(+) using LLaMA3/3.1-8B-Instruct, and compare it with a simple majority vote baseline. The results are presented in Table 2.

Table 2: Performance in out-of-distribution datasets

| AMC23 (LLaMA3.1) | | | | | | ADSIV (LLaMA3) | | | | | |
|---|---|---|---|---|---|---|---|---|---|---|---|
| DREAM(+) | | DREAM | | Majority Vote | | DREAM(+) | | DREAM | | Majority Vote | |
| acc | # tokens | acc | # tokens | acc | # tokens | acc | # tokens | acc | # tokens | acc | # tokens |
| 37.50% | 3186.38 | 30.00% | 3149.6 | 22.50% | 2851.45 | 95.35% | 160.55 | 93.02% | 162.65 | 93.02% | 193.93 |
| 47.50% | 6068.18 | 42.50% | 6256.20 | 22.59% | 5469.45 | 96.35% | 218.76 | 95.35% | 332.60 | 94.02% | 388.29 |
| 50.00% | 11210.10 | 47.50% | 12430.3 | 27.50% | 11708.9 | 97.67% | 543.02 | 96.01% | 650.10 | 95.35% | 782.80 |
| 60.00% | 23515.20 | 50.00% | 22971.6 | 30.00% | 22586.3 | 98.01% | 1112.1 | 97.34% | 1314.7 | 95.02% | 1558.1 |

From Table 2, we observe that across both datasets, DREAM with our reward model consistently achieves higher accuracy at the same level of tokens compared to majority vote, and DREAM(+) provide additional benefits in terms of accuracy-efficiency trade-off. Notably, on AMC23, which is significantly more challenging than the datasets used to train the reward model, our approach still provides strong guidance for intermediate reasoning, showing up to a 30% improvement over majority voting. These findings suggest that our reward model generalizes beyond its training distribution and demonstrates strong transferability to out-of-distribution math reasoning tasks.

## 4.4 Ablation Studies

In this section, we conduct ablation studies regarding the reward model size for math reasoning (Section 4.4.1), the effect of using the critic agent in code generation task, the synergy effect of DREAM and empirical and theoretical analysis on the selection of the thresholds. Due to space limitations, we defer the latter three studies to Appendix C.1, C.2 and C.3.

### 4.4.1 Reward Model Size

In this subsection, we study the impact of reward model size. In our main experiments for math reasoning, the reward model is fine-tuned from Qwen2.5-32B-Instruct, which is relatively large. To assess whether smaller models can serve as effective alternatives, we additionally fine-tune a reward model based on Qwen2.5-7B-Instruct. We then compare the performance of standard dual-phase search (without budget allocation) under two configurations: using Qwen2.5-7B-Instruct or Qwen2.5-32B-Instruct as the reward model. For reference, we also report results with simple majority voting. We conduct the experiments using LLaMA3/3.1-8B-Instruct as the reasoning models and the results are summarized in Table 3.

Table 3: Performance comparison across different reward model size.

| GSM8K | | | | | | MATH | | | | | |
|---|---|---|---|---|---|---|---|---|---|---|---|
| Qwen2.5-32B | | Qwen2.5-7B | | Majority vote | | Qwen2.5-32B | | Qwen2.5-7B | | Majority vote | |
| acc | # tokens | acc | # tokens | acc | # tokens | acc | # tokens | acc | # tokens | acc | # tokens |
| 91.43% | 601.13 | 87.34% | 606.191 | 83.40% | 646.18 | 61.00% | 2021.16 | 55.20% | 1829.73 | 41.00% | 1674.98 |
| 93.63% | 1221.21 | 93.63% | 1191.49 | 86.20% | 1297.45 | 66.80% | 3854.31 | 58.40% | 3576.18 | 46.00% | 3314.41 |
| 94.62% | 2463.82 | 94.62% | 2385.57 | 87.34% | 2598.09 | 70.80% | 7776.58 | 62.80% | 7477.92 | 50.00% | 6647.12 |
| 94.92% | 4916.89 | 94.92% | 4444.57 | 87.57% | 5202.77 | 72.60% | 15553.2 | 65.60% | 14557.8 | 51.40% | 13157.1 |

According to the results, the 32B reward model consistently outperforms the 7B version, achieving better accuracy-efficiency trade-offs across datasets. This suggests that, as larger instruction-tuned LLMs possess stronger intrinsic reasoning and representation capabilities, fine-tuning them with rollout-based supervision enables the reward function to better capture nuanced signals of correctness. Nevertheless, the 7B reward model also demonstrates strong effectiveness: although its step selection is not as precise as the 32B model, it still achieves significantly better accuracy–efficiency trade-offs compared to majority voting. This suggests that, with properly labeled training data, even moderately sized reward models can provide substantial benefits while reducing computations.

## 5 Conclusion

In this work, we proposed DREAM, a dual-phase test-time scaling framework that explicitly separates reasoning into planning and execution and conducts dedicated search in each phase. By equipping each phase with a reward model and introducing an adaptive budget allocation mechanism, our method enables finer-grained control over reasoning search, reduces wasted computation, and improves accuracy on both math reasoning and code generation tasks. Empirical results show that DREAM consistently outperforms standard beam search and prior PRM-based methods, highlighting the benefit of searching planning and execution separately and adaptively managing budget.

## ETHICS STATEMENT

We acknowledge the ICLR Code of Ethics and ensure that no concerns regarding the Code of Ethics arise from our work. Our study does not involve human subjects, personal or sensitive data, or experiments that could cause harm. All data used are either synthetic or publicly available under appropriate licenses, and we have adhered to principles of fairness, transparency, and reproducibility throughout.

## REPRODUCIBILITY STATEMENT

To ensure reproducibility, we provide implementation details and experimental settings in Section 4.1, and include additional information about hyperparameters and training configurations in Appendix B.

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

## A  APPENDIX

### A.1  ALGORITHM OF DREAM(+)

We present the detailed algorithm of DREAM(+) in Algorithm 1.

---

**Algorithm 1** Dual-Phase Search with budgets adjustment.

---

**Require:** Question $Q$, Planning/Execution Budget $N_1/N_2$, Thresholds $\tau_{p_1}/\tau_{p_2}$ and $\tau_{e_1}/\tau_{e_2}$, Beam Width $n_1/n_2$, Additional Budget limit $m_1/m_2$.

1: Initialize finished paths $F \leftarrow \emptyset$
2: Initialize step counter $s \leftarrow 1$
3: Initialize beam set $B \leftarrow \emptyset$
4: **while** $s \leq \max\_steps$ **do**
5:     **if** all $b \in B$ is finished **then**
6:         **break**
7:     **end if**
8:     $C_p(\text{candidates storage}) \leftarrow \emptyset$
9:     **if** $s = 1$ **then**
10:         Generate up to $N_1$ candidates for planning $p_1$ with reward scores $r$
11:         Early stop if there are $n_1$ planning all having reward $r > \tau_{p_1}$
12:         $C_p \leftarrow C_p \cup \{(p_1^{(i)}, r_1^{(i)}) \mid i = 1, \ldots, n\}$, $n$ denotes the number of actual sampling
13:         **if** all $r < \tau_{p2}$ **then**
14:             Generate up to additional $m_1$ candidates for planning $p_1$ with reward scores $r$
15:             Early stop if there are $n_1$ planning having $r > \tau_{p1}$
16:             $C_p \leftarrow C_p \cup \{(p_1^{(i)}, r_1^{(i)}) \mid i = 1, \ldots, n\}$
17:         **end if**
18:     **else**
19:         **for all** $b \in B$ **do**
20:             Generate up to $N_1/n_1$ candidates for planning $p_s$ with reward scores $r$
21:             Early stop if there are $N_1/n_1$ planning having $r > \tau_{p1}$
22:             $C_p \leftarrow C_p \cup \{(p_s^{(i)}, r_s^{(i)}) \mid i = 1, \ldots, n\}$
23:             **if** all $r < \tau_{p2}$ **then**
24:                 Generate up to $m_1$ additional candidates for planning $p_s$ with reward scores $r$
25:                 Early stop if there are $N_1/n_1$ planning having $r > \tau_{p1}$
26:                 $C_p \leftarrow C_p \cup \{(p_s^{(i)}, r_s^{(i)}) \mid i = 1, \ldots, n\}$
27:             **end if**
28:         **end for**
29:     **end if**
30:     Sort $C_p$ and take the top-$n_1$ planning $\{(p_1^1, e_1^1, \cdots, p_{s-1}^1, e_{s-1}^1, p_s^1, r_s^1), \cdots, (p_1^{n_1}, e_1^{n_1}, \cdots, p_{s-1}^{n_1}, e_{s-1}^{n_1}, p_s^{n_1}, r_s^{n_1})\}$
31:     Update beam $B \leftarrow \{(Q, p_1^1, e_1^1, \cdots, p_{s-1}^1, e_{s-1}^1, p_s^1), \cdots, (Q, p_1^{n_1}, e_1^{n_1}, \cdots, p_{s-1}^{n_1}, e_{s-1}^{n_1}, p_s^{n_1})\}$
32:     $C_e(\text{candidates storage}) \leftarrow \emptyset$
33:     **for all** $b \in B$ **do**
34:         Generate up to $N_2/n_2$ candidates for executions $e_s$ based on $p_s$ with reward scores $r'$
35:         Early stop if there are $N_2/n_2$ executions having reward $r' > \tau_{e1}$
36:         $C_e \leftarrow C_e \cup \{(p_s^{(i)}, e_s^{(i)}, r'_s^{(i)}) \mid i = 1, \ldots, n\}$, $n$ denotes the number of actual sampling
37:         **if** all $r' < \tau_2$ **then**
38:             Generate up to $m_2$ additional candidates for executions $e_s$ with reward scores $r'$
39:             Early stop if there are $N_2/n_2$ executions having reward $r' > \tau_{e1}$
40:             $C_e \leftarrow C_e \cup \{(p_s^{(i)}, e_s^{(i)}, r'_s^{(i)}) \mid i = 1, \ldots, n\}$
41:         **end if**
42:     **end for**
43:     Sort $C_e$ and take the top-$n_2$ executions $\{(p_1^1, e_1^1, \cdots, p_s^1, e_s^1, r'^1), \cdots, (p_1^{n_2}, e_1^{n_2}, \cdots, p_s^{n_2}, e_s^{n_2}, r'^{n_2})\}$
44:     Update beam $B \leftarrow \{(Q, p_1^1, e_1^1, \cdots, p_s^1, e_s^1), \cdots, (Q, p_1^{n_2}, e_1^{n_2}, \cdots, p_s^{n_2}, e_s^{n_2})\}$
45:     $s \leftarrow s + 1$
46: **end while**
47: **return** Path with highest reward in $B$

---

## A.2 EXAMPLES TO DEMONSTRATE DIFFICULTY VARIANTS IN REASONING STEPS.

In this subsection, we present real examples to illustrate difficulty diversity both across problems in a dataset and across steps within a single problem, motivating the need for dynamic allocation. Tables 4 and 5 provide examples from GSM8K and MATH: for each dataset, we include one easy problem, where intermediate steps have high average reward values and low variance, and one hard problem, where intermediate steps have lower average reward values and higher variance. For reference, we also provide the ground-truth solution for each example, offering a more intuitive sense of the difficulty of each problem. These examples demonstrate that difficulty variance arises not only across problems but also within individual reasoning trajectories.

Table 4: Examples of samples with different diversity in GSM8K datasets

---

Q: Josh decides to try flipping a house. He buys a house for \$80,000 and then puts in \$50,000 in repairs. This increased the value of the house by 150%. How much profit did he make?

Ground truth solution:
The cost of the house and repairs came out to $80,000 + 50,000 = 130,000$
He increased the value of the house by $80,000 * 1.5 = 120,000$
So the new value of the house is $120,000 + 80,000 = 200,000$
So he made a profit of $200,000 - 130,000 = 70000$
#### 70000

Average reward score for intermediate steps: 0.3613
Standard deviation of reward score: 0.184845

---

Q: A robe takes 2 bolts of blue fiber and half that much white fiber. How many bolts in total does it take?

Ground truth solution:
It takes $2/2 = 1$ bolt of white fiber
So the total amount of fabric is $2 + 1 = 3$ bolts of fabric
#### 3

Average reward score for intermediate steps: 0.9990
Standard deviation of reward score: 0.001746

---

Table 5: Examples of samples with different diversity in MATH datasets

---

Q: What is the smallest positive integer $n$ such that all the roots of $z^4 + z^2 + 1 = 0$ are $n^{\text{th}}$ roots of unity?

Ground Truth:
Multiplying the equation $z^4 + z^2 + 1 = 0$ by $z^2 - 1 = (z - 1)(z + 1)$, we get $z^6 - 1 = 0$.
Therefore, every root of $z^4 + z^2 + 1 = 0$ is a sixth root of unity.
The sixth roots of unity are $e^0$, $e^{2\pi i/6}$, $e^{4\pi i/6}$, $e^{6\pi i/6}$, $e^{8\pi i/6}$, and $e^{10\pi i/6}$.
We see that $e^0 = 1$ and $e^{6\pi i/6} = e^{\pi i} = -1$, so the roots of $z^4 + z^2 + 1 = 0$
are the remaining sixth roots of unity, namely $e^{2\pi i/6}$, $e^{4\pi i/6}$, $e^{8\pi i/6}$, and $e^{10\pi i/6}$.
The complex number $e^{2\pi i/6}$ is a primitive sixth root of unity, so by definition,
the smallest positive integer $n$ such that $(e^{2\pi i/6})^n = 1$ is 6.
Therefore, the smallest possible value of $n$ is $\boxed{6}$.

Average reward score for intermediate steps: 0.4206503
Standard deviation of the reward score: 0.3246094

---

Q: Simplify $\sqrt{242}$.

Ground truth:
Factor 242 as $11^2 \cdot 2$.
Then $\sqrt{242} = \sqrt{11^2} \cdot \sqrt{2} = \boxed{11\sqrt{2}}$.

Average reward score for intermediate steps: 0.9947
Standard deviation of reward score: 0.0024

---

Table 6: Candidate value of beam widths and sampling budgets

| $N_1$ | 2 | 4 | 8 | 16 | 32 |
|---|---|---|---|---|---|
| $N_2$ | 2 | 4 | 8 | 16 | 32 |
| $n_1$ | 1 | 2 | 2 | 4 | 4 |
| $n_2$ | 1 | 2 | 2 | 4 | 4 |

## B  ADDITIONAL DETAILS OF EXPERIMENTS SETTING

In this subsection, we provide additional details of experiment settings to ensure reproducibility.

### B.1  TRAINING AND INFERENCE CONFIGURATIONS

- For math reasoning, we develop the reward model by fine-tuning Qwen2.5-32B-Instruct for 2 epochs with a learning rate of $2.0 \times 10^{-6}$, using the Adam optimizer and a cosine learning-rate scheduler. The fine-tuning is conducted on 8 H100 GPUs with a batch size of 32. For code generation, we fine-tune Qwen2.5-Coder-7B-Instruct for 3 epochs, while keeping the other hyperparameters the same.

- During inference on math reasoning, we set the sampling temperature to 1.0 for LLaMA3/3.1-8B-Instruct and DeepSeek-MATH-7B-Instruct, and 0.5 for Qwen2.5-MATH-1.5B-Instruct.

- For code generation, we set the temperature to 1.0 across all experiments in DREAM and DREAM(+), while using 0.0 for CodeTree and Reflexion.

### B.2  SETTINGS OF THRESHOLD AND BUDGETS.

For GSM8K, we use $\tau_{p1} = \tau_{e1} = 0.99$, $\tau_{p2} = \tau_{e2} = 0.9$, and $m_1 = m_2 = 2$ for all models. For MATH, we set $\tau_{p1} = \tau_{e1} = 0.9$, $\tau_{p2} = \tau_{e2} = 0.5$, and $m_1 = m_2 = 2$. To control test-time scaling, we vary the sampling budget $N_1$, $N_2$ and the beam widths $n_1$, $n_2$. The candidate values used in our experiments are shown in Table 6; each column represents one configuration, corresponding to a single point in the accuracy–budget curves in the main results in Figure 4. Increasing these values proportionally increases inference-time computation.

Notably, we set $N_1 = N_2$ and $n_1 = n_2$ to keep the search space balanced between the planning and execution phases. Since both phases play important roles in producing a high-quality reasoning step, using symmetric widths prevents the search from being unintentionally biased toward one phase and also avoids introducing additional hyperparameter tuning.

## C  ADDITIONAL ABLATION STUDIES

### C.1  USAGE OF CRITIC AGENTS IN CODE GENERATION

In this subsection, we provide empirical evidence showing that when applying dual-phase search (DREAM) in the CodeTree method, the critic agent does not yield clear benefits in performance and, in fact, reduces efficiency. Table 7 presents results for DREAM+CodeTree under settings with and without critic agents. The accuracies on MBPP/HumanEval and MBPP+/HumanEval+ are reported as "weak acc" and "acc," respectively.

From the results, we observe that to achieve the same level of reasoning accuracy, DREAM with a critic agent consistently requires substantially more generated tokens. This indicates that, although the critic agent proposed by Li et al. (2024) was originally shown to improve accuracy given sufficient computation budget, its efficiency is inferior to simply scaling generation multiple times and directly applying the percentage of passed test cases as the reward signal. These findings support our design choice of removing the critic agent from the code generation framework, simplifying the system while preserving performance and improving efficiency.

Table 7: Comparison of performance with/without critic agent

| MBPP(+) | | | | | | HumanEval(+) | | | | | |
|---|---|---|---|---|---|---|---|---|---|---|---|
| without critic | | | with critic | | | without critic | | | with critic | | |
| weak acc | acc | # tokens | weak acc | acc | # tokens | weak acc | acc | # tokens | weak acc | acc | # tokens |
| 81.49% | 68.52% | 569.59 | 81.49% | 68.52% | 1051.52 | 75.61% | 67.69% | 721.28 | 75.61% | 67.69% | 1235.92 |
| 86.51% | 74.10% | 771.66 | 87.58% | 74.88% | 1348.16 | 81.71% | 72.57% | 1205.10 | 78.66% | 70.13% | 1596.59 |
| 88.37% | 74.09% | 976.72 | 86.25% | 73.02% | 1627.35 | 84.14% | 72.57% | 1619.51 | 80.49% | 71.34% | 1827.49 |
| 90.48% | 75.14% | 1292.83 | 88.37% | 76.19% | 2074.12 | 84.14% | 75.00% | 2307.87 | 81.71% | 71.95% | 2311.83 |
| 90.21% | 76.98% | 2081.73 | 90.73% | 75.94% | 2620.38 | 87.80% | 78.04% | 4552.75 | 87.20% | 76.83% | 4833.02 |

## C.2  SYNERGY EFFECT OF DREAM

In this subsection, we demonstrate the synergy between dual-phase search and dynamic budget allocation. Specifically, we compare the benefits of applying budget allocation to dual-phase search versus standard beam search. Figure 6 plots the accuracy–tokens frontier of four methods: DREAM, DREAM(+), Beam Search, and Beam Search(+), where the "+" variants denote the incorporation of dynamic budget allocation. From the figure, we observe that budget allocation also improves the accuracy–efficiency trade-off when applied to standard beam search, indicating the general effectiveness of this design. However, the gains for Beam Search(+) are consistently smaller than those achieved by DREAM(+). This comparison highlights the complementary nature of the two components: dual-phase search and dynamic budget allocation. The synergy arises because dual-phase search reduces wasted computation by separating planning and execution, while dynamic budget allocation further reallocates saved resources to harder steps, making the two mechanisms mutually reinforcing.

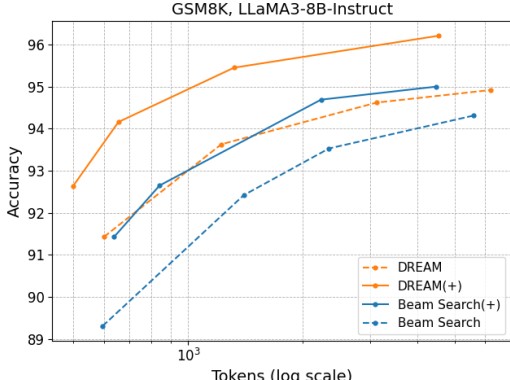

Figure 6: Accuracy-Tokens Frontier across methods.

## C.3  ABLATION STUDIES ON THE THRESHOLDS.

In this subsection, we provide (1). Justification about why $\tau_{e_1} = \tau_{p_1}$; and (2). Theoretical and empirical analysis on how different choices of the thresholds $\tau_{e1}, \tau_{p1}, \tau_{e2}, \tau_{p2}$ affect the performance of DREAM+.

### C.3.1  JUSTIFICATION ABOUT WHY $\tau_{e_1} = \tau_{p_1}$

To justify why $\tau_{e_1} = \tau_{p_1}$, empirically, as shown in Table 8, using the same threshold ($\tau_{e_1} = \tau_{p_1}$) yields slightly better performance.

The intuition behind this observation is twofold. Firstly, although planning and execution operate on different sampling spaces, their reward scores share a comparable numerical scale: both reward models evaluate step-level reasoning quality (whether a subgoal is sensible, or whether a subexecution is correct). Since the rewards are trained under the same supervision paradigm, applying the same threshold yields consistent pruning behavior in both phases.

Table 8: Performance of DREAM in gsm8k with Qwen2.5-MATH-1.5B-Instruct when $\tau$s take different values. (We set $N_1 = N_2 = 4$, $n_1 = n_2 = 2$)

| $\tau_2 = 0.7$ | | | $\tau_2 = 0.9$ | | |
|---|---|---|---|---|---|
| $\tau_{p_1}/\tau_{e_1}$ | **acc** | **tokens** | $\tau_{p_1}/\tau_{e_1}$ | **acc** | **tokens** |
| 0.9/0.8 | 89.40% | 501.276 | 0.99/0.9 | 90.00% | 569.846 |
| 0.8/0.9 | 89.00% | 492.466 | 0.9/0.99 | 91.20% | 525.868 |
| 0.85/0.85 | 90.00% | 492.182 | 0.95/0.95 | 91.60% | 522.408 |

Table 9: Performance of DREAM+ with different values of thresholds (GSM8k, Qwen2.5-MATH-1.5B-Instruct)

| $\tau_{e_1} = \tau_{p_1} = 0.99$ | | | $\tau_{e_2} = \tau_{p_2} = 0.3$ | | |
|---|---|---|---|---|---|
| $\tau_{e_2}/\tau_{p_2}$ | **acc** | **tokens** | $\tau_{e_1}/\tau_{p_1}$ | **acc** | **tokens** |
| 0.9/0.9 | 0.92 | 538.754 | 0.99/0.99 | 0.892 | 489.742 |
| 0.7/0.7 | 0.902 | 520.430 | 0.8/0.8 | 0.884 | 472.432 |
| 0.5/0.5 | 0.896 | 506.496 | 0.6/0.6 | 0.878 | 467.584 |
| 0.3/0.3 | 0.892 | 489.742 | 0.4/0.4 | 0.876 | 465.156 |

Secondly, Different thresholds can easily create imbalanced pruning: Both planning and execution are essential for producing a good reasoning step. If one phase is pruned more aggressively than the other, the search becomes unbalanced. In such cases, either valid plans are discarded too early, or promising plans cannot be properly expanded. Therefore, using the same threshold keeps the pruning strength aligned across phases and prevents either side from dominating the search.

### C.4 THEORETICAL AND EMPIRICAL ANALYSIS ON THE INFLUENCE OF THE THRESHOLDS.

**Empirical Results.** In Table 9, we provide empirical results on how different values of $\tau_{e1}, \tau_{p1}, \tau_{e2}, \tau_{p2}$ influence the performance of DREAM+.

The left block reports results under a fixed high early step threshold ($\tau_{e1} = \tau_{p1} = 0.99$) while varying the extra-budget thresholds $\tau_{e2}/\tau_{p2}$. The right block reports results under a fixed low extra-budget thresholds ($\tau_{e2} = \tau_{p2} = 0.3$) while varying the early stop thresholds $\tau_{e1}/\tau_{p1}$.

From the results in Table 9, we observe two consistent trends.

First, when $\tau_1$ is fixed, decreasing $\tau_2$ leads to lower accuracy and fewer generated tokens. This result is expected, as a higher $\tau_2$ imposes a stricter extra budget standard, causing the model to more frequently determine that the current candidates are not sufficient and therefore allocate more budget to explore further. As $\tau_2$ decreases, fewer steps trigger budget expansion, reducing exploration and slightly lowering accuracy.

Second, when $\tau_2$ is fixed, decreasing $\tau_1$ also reduces accuracy and the number of generated tokens. This is because the thresholds $\tau_1$ determine whether the remaining budget for a step should continue to be used. A higher $\tau_1$ corresponds to a stricter stopping criterion and encourages generating more candidates within the step, thereby resulting in higher accuracy and higher computation costs.

Overall, $\tau_1$ and $\tau_2$ control different aspects of intra-step search, and their effects on accuracy and efficiency follow intuitive patterns.

**Theoretical Analysis: Markov-based Analysis of the Threshold Behaviors.**

To build a theoretical understanding of how the thresholds $\tau_1$ and $\tau_2$ influence DREAM+, we begin by modeling the within-step search as a simplified Markov process. To begin, we model the single step generation in DREAM+ as a simple Markov process with states $\{1, 2, 3, ..., n, *\}$ where state $*$ denotes the end state. For any states $a$ and $b$, let $P(x_{i+1} = a \mid x_i = b) = P_{b \to a}$. We adopt the following assumptions for the transition matrix.

1. $P_{b \to b+1} = \frac{1}{2}$ for any $1 \le b < n$

2. $P_{a \to *} = \frac{1}{2}$ for any $1 \le a \le n$

3. $P_{n \to n} = \frac{1}{2}$

At step $i$, each beam samples multiple candidates for next states and selects the one with the highest reward value based on the reward model. This process can be modeled as: at state $x_i = b$, we draw several independent samples of $x_{i+1}$ under a sampling budget $k$, and retain the candidate with the highest reward. We further assume that the reward assigned by the reward model is positively correlated with the transition likelihood toward the end state $*$, $P^\infty_{x_{i+1} \to *}$. Then we have the following theorem:

**Theorem C.1** *Under the assumptions described above, suppose the maximum number of reasoning steps is $M$ ($M$ is sufficiently large). Then starting from state $b$, the probability of eventually reaching the terminal state $*$ within the remaining $M - i$ steps is: $P^{(M-i)}_{b \to *} = 1 - 2^{-(M-i-b+1)}$. Furthermore, if at step $i$ we sample $k$ candidates for $x_{i+1}$ and keep the one with the highest reward, then the probability of successfully reaching the end state becomes: $P^* = 1 - (\frac{1}{2^{M-i-b+1}})^k$.*

Based on the above threshold, we have the following two propositions on the impact of $\tau_1$ and $\tau_2$ towards $P^*$. For simplicity, each proposition considers $\tau_1$ and $\tau_2$ respectively.

**Proposition C.1** *Let $p = 1 - \frac{1}{2^{(M-i-b)}}$. Assume the sampling budget is and $\tau_2 = 0$. Then:*
*(1) If $\tau_1 \in (0, p]$, the probability of successfully reaching the end states is: $P^* = 1 - (\frac{1}{2^{M-i-b+1}})$.*
*(2) If $\tau_1 \in (p, 1]$, $P^* = 1 - (\frac{1}{2^{M-i-b+1}})^k$*
*When $k > 1$, $P^*$ is higher when $\tau_1 \in (p, 1]$.*

**Proposition C.2** *Let $p = 1 - \frac{1}{2^{(M-i-b)}}$. Assume that the original sampling budget is $k$, the maximum additional budget triggered by $\tau_2$ is $k'$, and $\tau_1 = 1$ is fixed. Then:*
*(1) If $\tau_2 \in (0, p]$, the probability of successfully reaching the end states is: $P^* = 1 - (\frac{1}{2^{M-i-b+1}})^k$.*
*(2) If $\tau_2 \in (p, 1]$, $P^* = 1 - (\frac{1}{2^{M-i-b+1}})^{(k+k')}$*
*Thus, $P^*$ is higher when $\tau_2 \in (p, 1]$.*

The two proposition provide evidence that, when all the other conditions are fixed, increasing either $\tau_1$ or $\tau_2$ increase the probability of successfully reaching the end state increase. This theoretical analysis explains why larger thresholds generally lead to better performance in DREAM+, as they effectively allocate more sampling budget to promising reasoning paths.
For simplicity, our analysis considers the effect of each threshold independently by fixing one and varying the other. Intuitively, increasing both $\tau_1$ and $\tau_2$ at the same time would compound their effects, which tightening selection and increasing the effective sampling budget, thereby further enhancing the probability of reaching the correct solution.

# D  ADDITIONAL EVALUATION

## D.1  EXPERIMENTS IN MORE CHALLENGING DATASETS AND MORE CAPABLE MODELS

To provide further empirical evidence for the effectiveness of our method, in this subsection we additionally evaluate Qwen2.5-MATH-72B (Yang et al., 2024b), one of the strongest math-reasoning models, on two harder benchmarks: AMC23 (Math-AI, 2023) and GSM8K-Hard (Gao et al., 2023). In these experiments, we use the same reward model as the one used in the experiments of Section 4.2.1. The results are presented in Table 12 and 11 and plot in Figure 8. From the results, we can observe that, DREAM consistently achieves a better accuracy–efficiency trade-off across both datasets compared with the baselines, demonstrating that the benefits of our method generalize to stronger models and more difficult reasoning tasks.

### D.2 COMPARISON BETWEEN PLANNING/EXECUTION DECOMPOSITION AND INCREASING GRANULARITY OF CoT

In this subsection, we provide comparison between planning/execution decomposition and increasing granularity of CoT to further demonstrate that the effectiveness brought by our method does not come from simply making the reasoning steps more fine-grained, but from the structural decomposition that separates high-level subgoal planning from low-level execution.

To begin with, we would like to highlight the fundamental difference between plan–execution decomposition and simply increasing step granularity. Specifically, increasing granularity means dividing a single reasoning step into smaller sequential segments (e.g., splitting one step into two micro-steps). In contrast, plan–execution decomposition separates two different types of reasoning behaviors. The planning component focuses on forming high-level strategic decisions or subgoals, while the execution component focuses on concretely carrying out these decisions. Consequently, decomposing a step into plan and execution changes the structure of the reasoning process rather than merely adjusting the level of granularity. To better illustrate the difference, we provide real examples demonstrating both strategies and the combination of both strategies:

Q: Jack deposited $4000 in a bank with a 1% annual interest rate. What will be the total amount after one year?

**1. Standard CoT:**
Step 1: With 1% interest rate, after the first year, the total value is 10000×(1+1%)=10100
Step 2: With 2% interest rate, after the second year, the total value is 10100(1+2%)=10302

**2. CoT with increased granularity:**
Step 1: With 1% interest rate, the first y ear's interest is 10000*1%=100.
Step 2: The total value after the first year is 10000+100=10100.
Step 3: With 2% interest rate, the second year's interest 10100*2% = 202.
Step 4: The total value after the second year is 10100+202=10302.

**3. CoT with plan/execution decomposition:**
Step 1: Determine the total value after the first year:
- Given that the interest rate of the 1st year is 1%. The total value becomes 10000×(1+1%)=10100
Step 2: Determine the total value after the second year:
- Given that the interest rate of the 1st year is 2%. The total value becomes 10100×(1+2%)=10302.

**4. CoT with plan/execution decomposition + increased granularity:**
Step 1: Determine the first year's interest:
- Given that the interest rate of the 1st year is 1%. The first year's interest is 1% x 10000=100
Step 2: Determine the total value after the first year:
- After the first year, the interest is 10000+100=10100
Step 3: Determine the second year's interest:
- Given that the interest rate of the 2nd year is 2%. The second year's interest is 2% x 10100=202
Step 4: Determine the total value after the second year:
- After the second year, the total value becomes 10100+202 = 10302.

To better demonstrate that the improvement of DREAM comes from structural plan–execution decomposition rather than simply splitting steps into smaller units, we compare all the four CoT variants shown in the above example using DREAM under different beam widths. The numerical results are provided in Table 10 and visualized in Figure 7.

Based on the results, we have the following observations. First, when comparing entries within each row of the table i.e., under the same searching width, we observe both increasing granularity and applying plan–execution decomposition can improve accuracy, but both approaches also require more tokens and therefore higher computation. However, the accuracy–efficiency trade-off shown in Figure 1 reveals a clear difference: there is no obvious gain from finer-grained steps in terms of accuracy-efficiency trade-off, whereas the gain from plan–execution decomposition is obvious.

Second, when the beam width is fixed, combining increased granularity with plan–execution decomposition leads to higher accuracy than using plan–execution alone. However, this combined strategy substantially increases the number of generated tokens and makes the reasoning trace significantly

Table 10: Comparison of four CoT formats on the MATH dataset with LLaMA3.1-8B-Instruct

| Standard | | Granularity | | Plan / Execution | | Plan/Execution + Granularity | |
|---|---|---|---|---|---|---|---|
| acc | tokens | acc | tokens | acc | tokens | acc | tokens |
| 0.480 | 811.938 | 0.484 | 1011.53 | 0.522 | 949.49 | 0.538 | 1302.44 |
| 0.568 | 1578.09 | 0.570 | 1902.53 | 0.610 | 2021.16 | 0.614 | 2631.79 |
| 0.610 | 3170.16 | 0.638 | 3805.05 | 0.668 | 3854.31 | 0.650 | 5149.69 |
| 0.658 | 6188.42 | 0.694 | 7872.97 | 0.708 | 7776.57 | 0.714 | 10587.93 |

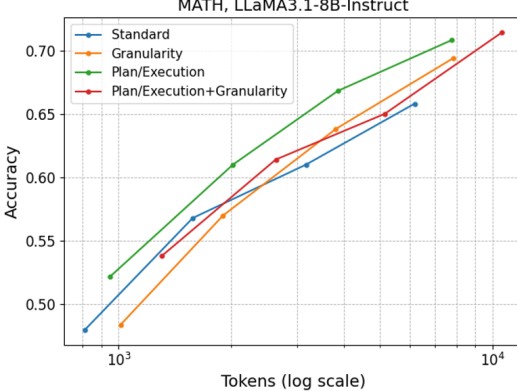

Figure 7: Comparison of accuracy-efficiency trade-off of the four CoT formats

more verbose. As a consequence, the computational cost grows faster than the improvement in accuracy. When examining the accuracy–efficiency trade-off, we do not observe advantages beyond using plan–execution decomposition alone.

In summary, the results confirm that the benefit of DREAM does not come from simply making the reasoning steps more fine-grained, but from the structural decomposition that separates high-level subgoal planning from low-level execution, enabling more effective exploration and pruning and ultimately yielding a better accuracy–efficiency trade-off.

# E   FEW-SHOT PROMPTS FOR GUIDING REASONING IN THE PLAN–EXECUTION FORMAT

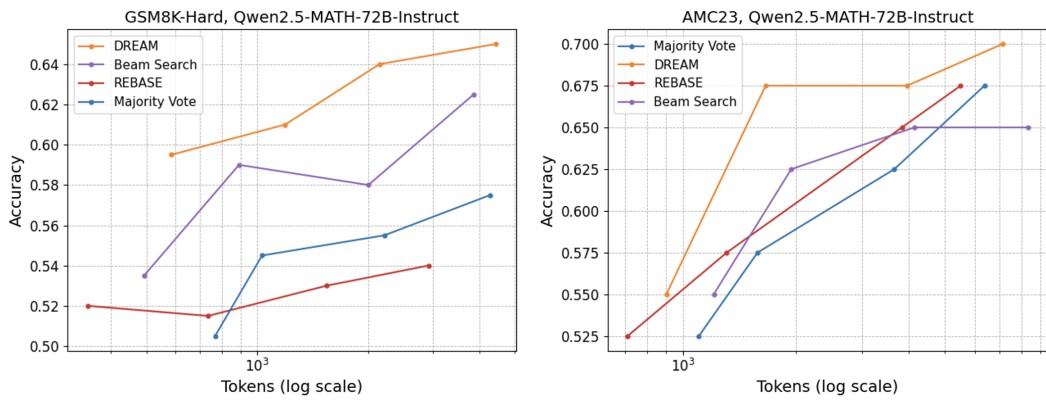

Figure 8: Accuracy-Efficiency Trade-off on Hard Datasets with Qwen2.5-MATH-72B-Instruct

Table 11: Qwen2.5-MATH-72B-Instruct on GSM8k-Hard

| DREAM | | Beam | | Rebase | | Majority Vote | |
|---|---|---|---|---|---|---|---|
| acc | tokens | acc | tokens | acc | tokens | acc | tokens |
| 0.595 | 584.345 | 0.535 | 492.910 | 0.5200 | 347.075 | 0.505 | 768.035 |
| 0.610 | 1190.035 | 0.590 | 891.890 | 0.5150 | 735.220 | 0.545 | 1030.260 |
| 0.640 | 2146.050 | 0.580 | 2001.450 | 0.530 | 1541.035 | 0.555 | 2210.765 |
| 0.650 | 4436.815 | 0.625 | 3863.665 | 0.540 | 2911.175 | 0.575 | 4281.115 |

Table 12: Qwen2.5-MATH-72B-Instruct on AMC23

| DREAM | | Beam Search | | REBASE | | Majority Vote | |
|---|---|---|---|---|---|---|---|
| acc | tokens | acc | tokens | acc | tokens | acc | tokens |
| 0.55 | 901.7 | 0.550 | 1206.625 | 0.525 | 710.425 | 0.525 | 1101.425 |
| 0.675 | 1660.2 | 0.625 | 1943.300 | 0.575 | 1306.475 | 0.575 | 1579.400 |
| 0.675 | 3960.83 | 0.650 | 4142.125 | 0.650 | 3837.675 | 0.625 | 3663.275 |
| 0.700 | 7128.45 | 0.650 | 8314.875 | 0.675 | 5503.650 | 0.675 | 6373.825 |

Table 13: Few-shot Prompt for the GSM8K dataset

---

Learn from the following examples to answer the given question. For each step, begin by stating the general plan, followed by executing that plan (started with "-"), separated by a colon. When the answer is finished, end with "The final answer is:".

Q: A father is building a playset for his son and needs to purchase lumber, nails, and fabric. When he started planning the project, the necessary lumber cost $450, the nails cost $30, and the fabric cost $80. However, recent economic inflation has caused the price of lumber to increase by 20%, the price of nails to increase by 10%, and the price of fabric to increase by 5%. In dollars, how much more money will it cost to complete the project now (after inflation) than it would have when the father began planning?
A: Let's think step by step.
Step 1: Find the new cost of lumber:
- The lumber originally cost $450, and with a 20% increase it becomes $450 \times 1.20 = 540$$.
Step 2: Find the new cost of nails:
- The nails originally cost $30, and with a 10% increase they become $30 \times 1.10 = 33$.
Step 3: Find the new cost of fabric:
- The fabric originally cost $80, and with a 5% increase it becomes $80 \times 1.05 = 84$.
Step 4: Find the total original cost:
- Before inflation, the total was $450 + 30 + 80 = 560$.
Step 5: Find the total new cost:
- After inflation, the total is $540 + 33 + 84 = 657$.
Step 6: Compare the costs:
- The difference between the new and old total is $657 - 560 = 97$. The final answer is: 97.

Q: There are five times as many swordfish as pufferfish in an aquarium exhibit. If there are 90 fish total, how many pufferfish are there?
A: Let's think step by step.
Step 1: Express swordfish in terms of pufferfish:
- If the number of pufferfish is $x$, then the number of swordfish is $5x$.
Step 2: Write the total number of fish:
- The total is $x + 5x = 6x$.
Step 3: Set up the equation:
- Since the total is 90, we have $6x = 90$.
Step 4: Solve for $x$:
- Dividing both sides by 6 gives $x = 15$. The final answer is: 15.

Q: Mrs. Johnson's class raised $2300, which is twice what Mrs. Sutton's class raised. If Mrs. Sutton's class raised 8 times less than Miss Rollin's class, and Miss Rollin's class raised one-third of the total, how much money did the school raise after deducting 2% fees?
A: Let's think step by step.
Step 1: Find Mrs. Sutton's amount:
- Since Mrs. Johnson's $2300 is twice Sutton's, Sutton's amount is $2300 / 2 = 1150$$.
Step 2: Find Miss Rollin's amount:
- Rollin's class raised 8 times more than Sutton's, so $1150 \times 8 = 9200$.
Step 3: Find the school's total before fees:
- Since Rollin's amount is one-third of the total, the total is $9200 \times 3 = 27600$.
Step 4: Calculate the deduction:
- 2% of 27600 is $27600 \times 0.02 = 552$.
Step 5: Find the net amount:
- Subtracting the deduction gives $27600 - 552 = 27048$. The final answer is: 27048.

Q: John buys 500 newspapers. Each newspaper sells for $2. He sells 80% of them. He buys them all for 75% less than the price at which he sells them. How much profit does he make?
A: Let's think step by step.
Step 1: Determine how many he sells:
- He sells 80% of 500, which is $0.8 \times 500 = 400$.
Step 2: Determine the selling price:
- Each newspaper sells for $2.
Step 3: Compute revenue:
- Selling 400 at $2 each gives $400 \times 2 = 800$.
Step 4: Determine purchase price per paper:
- He bought them 75% cheaper, meaning at 25% of 2, which is $0.50$.
Step 5: Compute total cost:
- Buying 500 at $0.50 each costs $500 \times 0.50 = 250$.
Step 6: Calculate profit:
- Revenue minus cost is $800 - 250 = 550$. The final answer is: 550.

Continue:

Q: A house's living room, dining room and kitchen take up 1,000 sq ft. The house also has a guest bedroom and a master bedroom suite. If the guest bedroom is a quarter of the size of the master bedroom suite, and the house is 2,300 sq ft total, how large is the master bedroom suite?
A: Let's think step by step.
Step 1: Find how much space is left for the bedrooms:
- The house is 2,300 sq ft total, and 1,000 sq ft is already used by the living room,
dining room, and kitchen. So the bedrooms must occupy $2300 - 1000 = 1300$ sq ft.
Step 2: Express the guest bedroom in terms of the master bedroom:
- If the master bedroom is $x$ sq ft, the guest bedroom is one quarter of that, or $x/4$.
Step 3: Write the total space of both bedrooms:
- Together the master and guest bedrooms occupy $x + x/4 = (5/4)x$.
Step 4: Solve for the master bedroom size:
- Since the two bedrooms together are 1,300 sq ft, we have $(5/4)x = 1300$.
Solving gives $x = (1300 \times 4)/5 = 1040$. The final answer is: 1040.

Q: Carter usually bakes 6 cheesecakes, 5 muffins, and 8 red velvet cakes regularly for a week. For this week he was able to bake triple the number of cheesecakes,muffins, chocolate moist cakes, and red velvet cakes. How much more cakes was Carter able to bake for this week?
A: Let's think step by step.
Step 1: Calculate Carter's usual total:
- Normally he bakes $6 + 5 + 8 = 19$ cakes per week.
Step 2: Find his cheesecakes this week:
- Tripling 6 cheesecakes gives $6 \times 3 = 18$.
Step 3: Find his muffins this week:
- Tripling 5 muffins gives $5 \times 3 = 15$.
Step 4: Find his red velvet cakes this week:
- Tripling 8 red velvet cakes gives $8 \times 3 = 24$.
Step 5: Add up the new totals:
- Altogether he baked $18 + 15 + 24 = 57$ cakes this week.
Step 6: Compare to his usual:
- Compared to 19 cakes normally, the increase is $57 - 19 = 38$. The final answer is: 38.

Q: Four adults with 32 teeth went to the dentist. The first had 1/4 removed, the second 3/8 removed, the third half removed, and the fourth 4 removed. What's the total number of teeth removed?
A: Let's think step by step.
Step 1: Find the first person's removal:
- One-quarter of 32 teeth is $32 \times 1/4 = 8$.
Step 2: Find the second person's removal:
- Three-eighths of 32 teeth is $32 \times 3/8 = 12$.
Step 3: Find the third person's removal:
- Half of 32 teeth is $32 \times 1/2 = 16$.
Step 4: Note the fourth person's removal:
- This was directly stated as 4 teeth.
Step 5: Add all removals together:
- $8 + 12 + 16 + 4 = 40$ teeth removed in total. The final answer is: 40.

Q: Joy fosters dogs. The mom dog eats 1.5 cups of food three times a day. The puppies (5 total) each eat 0.5 cups twice a day. How much food will Joy need for the next 6 days?
A: Let's think step by step.
Step 1: Calculate the mom's daily food:
- She eats 1.5 cups at each of 3 meals, so $1.5 \times 3 = 4.5$ cups per day.
Step 2: Calculate one puppy's daily food:
- Each puppy eats 0.5 cups at 2 meals, so $0.5 \times 2 = 1$ cup per day.
Step 3: Calculate all puppies' daily food:
- With 5 puppies, their total is $1 \times 5 = 5$ cups per day.
Step 4: Find the combined daily food:
- Adding the mom and puppies gives $4.5 + 5 = 9.5$ cups per day.
Step 5: Find the total for 6 days: Over 6 days,
- Joy needs $9.5 \times 6 = 57$ cups. The final answer is: 57.

Q: [New Question to be Solved]

Table 14: Few-show prompt for the MATH dataset

Learn from the following examples to answer the given question. For each step, begin by stating the general plan, followed by executing that plan (started with "-"), separated by a colon. When the answer is finished, end with "The final answer is:".

Q: Karl was attempting to calculate economic figures. He found the following equation to be true: $fp - w = 10000$. If $f = 5$ and $w = 5 + 125i$, what is $p$?
A: Step 1: Write down the expression for fp in terms of w:
- $fp = 10000 + w$.
Step 2: Write down the expression for fp using the value of $w$:
- Substitute $w = 5 + 125i$ into the equation, we get $fp = 10000 + 5 + 125i = 10005 + 125i$.
Step 3: Calculate the value of $p$:
- Since $f = 5$, we compute $p = \frac{fp}{f} = \frac{10005 + 125i}{5} = 2001 + 25i$. The final answer is: $2001 + 25i$.

Q: The roots of the equation $2x^2 - mx + n = 0$ sum to 6 and multiply to 10.
What is the value of $m + n$?
A: Step 1: Identify the relationships between the roots and coefficients of a quadratic:
- For a quadratic $ax^2 + bx + c = 0$, the sum of the roots is $-\frac{b}{a}$ and the product is $\frac{c}{a}$.
Step 2: Use the sum of roots to find $m$:
- Here, $a = 2$ and $b = -m$, so the sum is $-\frac{-m}{2} = \frac{m}{2}$.
Setting this equal to 6 gives $\frac{m}{2} = 6 \Rightarrow m = 12$. Step 3: Use the product of roots to find $n$:
- The product is $\frac{n}{2}$, and it's given as 10. So $\frac{n}{2} = 10 \Rightarrow n = 20$.
Step 4: Add $m$ and $n$:
- $m + n = 12 + 20 = 32$. The final answer is: 32.

Question: If the two roots of the quadratic $7x^2 + 3x + k$ are $\frac{-3 \pm i\sqrt{299}}{14}$, what is $k$?
A: Step 1: Recall the quadratic formula:
- The roots of $ax^2 + bx + c = 0$ are given by $\frac{-b \pm \sqrt{b^2 - 4ac}}{2a}$.
Step 2: Identify coefficients:
- From the equation $7x^2 + 3x + k$, we have $a = 7$, $b = 3$, $c = k$.
Step 3: Write the discriminant in terms of $k$:
- $b^2 - 4ac = 3^2 - 4 \cdot 7 \cdot k = 9 - 28k$.
Step 4: Use the root form to determine the discriminant:
- Since the roots are $\frac{-3 \pm i\sqrt{299}}{14}$, the discriminant is $-299$. So $9 - 28k = -299 \Rightarrow 28k = 308 \Rightarrow k = 11$.
The final answer is: 11.

Question: Rationalize the denominator of $\frac{1}{\sqrt{2} + \sqrt{3} + \sqrt{7}}$, and write your answer in the form $\frac{A\sqrt{2} + B\sqrt{3} + C\sqrt{7} + D\sqrt{E}}{F}$. What is $A + B + C + D + E + F$?
A: Step 1: Multiply by a conjugate to simplify:
- Multiply numerator and denominator by $(\sqrt{2} + \sqrt{3}) - \sqrt{7}$.
Step 2: Simplify the denominator:
- $(\sqrt{2} + \sqrt{3})^2 = 2 + 3 + 2\sqrt{6} = 5 + 2\sqrt{6}$, and the full denominator becomes $(5 + 2\sqrt{6}) - 7 = -2 + 2\sqrt{6}$.
Step 3: Multiply by the conjugate of the new denominator:
- Multiply numerator and denominator by $-2 - 2\sqrt{6}$: $\frac{\sqrt{2} + \sqrt{3} - \sqrt{7}}{-2 + 2\sqrt{6}} \cdot \frac{-2 - 2\sqrt{6}}{-2 - 2\sqrt{6}} = \frac{4\sqrt{2} + 3\sqrt{3} - \sqrt{7} - \sqrt{42}}{10}$.
Step 4: Identify constants and sum:
- $A = 4, B = 3, C = -1, D = -1, E = 42, F = 10$,
so the total is: $A + B + C + D + E + F = 4 + 3 - 1 - 1 + 42 + 10 = 57$. The final answer is: 57.

Question: For what values of $b$ is $-2$ not in the range of the function $f(x) = x^2 + bx + 2$?
Express your answer in interval notation.
A: Step 1: Understand when $-2$ is in the range:
- $-2$ is in the range if there exists some $x$ such that $f(x) = -2$.
Step 2: Set up the equation:
- $f(x) = -2 \Rightarrow x^2 + bx + 2 = -2 \Rightarrow x^2 + bx + 4 = 0$.
Step 3: Determine when the equation has no real solutions:
- A quadratic has no real solutions when the discriminant is negative.
Step 4: Solve for values of $b$:
- $b^2 - 16 < 0 \Rightarrow b^2 < 16 \Rightarrow -4 < b < 4$. The final answer is: $(-4, 4)$.

Continue:

Question: What is the value of $\sqrt[3]{3^5 + 3^5 + 3^5}$?
A: Step 1: Calculate $3^5$:
- $3^5 = 243$.
Step 2: Add three copies of $3^5$:
- $3^5 + 3^5 + 3^5 = 243 + 243 + 243 = 729$.
Step 3: Take the cube root:
- $\sqrt[3]{729} = 9$ since $9^3 = 729$.
The final answer is: 9.

Q: [New Question to be Solved]

