# OpenReview forum: "Adaptive Test-Time Reasoning via Reward-Guided Dual-Phase Search"
_ICLR.cc/2026/Conference — ICLR 2026 Conference Withdrawn Submission_

### Official Review · Reviewer_B7cR · 2025-10-24

**Soundness:** 2
**Presentation:** 2
**Contribution:** 2
**Rating:** 2
**Confidence:** 4

**Summary:**

This work explicitly decompose the tree-based search steps into planning and execution phases, and use a reward model to guide pruning the selection. It also introduces a dynamic budget allocation mechanism to allocate the sampling effort for each search step. Experimental results demonstrate consistent improvement on both math and coding tasks.

**Strengths:**

1. The proposed method sounds reasonable for the reasoning tasks
2. Comprehensive experiments are performed, with various tasks and base models, at a wide range of token budgets. The results are also promising.

**Weaknesses:**

1. The main issue lies in the lack of technical contribution. In short, the only novel part of this work lies in the proposal to explicitly split the search step into planning and decomposition phases. But why it would work, what is the nature of planning and execution from the machine learning perspective, how the paradigm could be generalized to other tasks, seem to be unclear. The authors leave all these questions unanswered, leaving very few takeaways from the reader side.
2. The proposed method contains a bag of tricks, but the ablation study does not perform a careful evaluation to verify which component makes the main contribution.
3. There are too many heuristic designs, such as the width and budgets. All these factors make the proposed method hard to generalize.

**Questions:**

1. How the model would behave when no explicitly decomposing the search into planning and execution (paradigm Figure 3(c))?
2. How do you instruct the model to do planning and execution? Have you trained the model to do it?

---

> ### Author Response · Authors · 2025-11-22
> **Response to Reviewer B7cR (1/5)**
>
> We appreciate your efforts in reviewing our paper and recognizing our contribution in the solid theories. We will revise our paper based on your suggestions, and below are some answers to your comments and questions:
>
> > Weakness 1:  (1) Why explicitly spliting the search step into planning and decomposition phases would work, (2) What is the nature of planning and execution from the machine learning perspective, (3) How the paradigm could be generalized to other tasks, seem to be unclear.
>
> **Response:** We thank the reviewer for raising the questions. In the following, we explain them one by one:
>
> (1) Why splitting into plan/execution work?
>
> Splitting reasoning into planning and execution works because these two stages correspond to different types of reasoning. Planning focuses on high-level structural decisions, such as identifying subgoals or decomposition strategies, while execution focuses on low-level computations and concrete derivations. Their error modes are different: planning errors are semantic and directional, whereas execution errors are typically arithmetic or local.
>
> Given these distinct roles and error modes, separating the two stages allows reward models to provide phase-specific supervision: PRM_plan assesses strategic soundness, and PRM_exec evaluates step-level correctness, leading to better utilization of reward signals. In addition, by separating them, weak plans can be discarded early, while promising plans can be paired with multiple execution attempts, significantly reducing search waste. Overall, this hierarchical structure improves both accuracy and efficiency by aligning computation and reward guidance with the inherent structure of reasoning.
>
> (2) What is the nature of planning and execution from the machine learning perspective?
>
> From a machine-learning perspective, following a similar view of [1], the two phases of reasoning in our approach, planning and execution, can be viewed as the hidden and observed states in a Hidden Markov Model (HMM). Specifically, planning involves choosing high-level structural decisions (such as subgoals or decomposition strategies) that determine the global direction of reasoning and serve as the structure of the solution; these decisions therefore play the role of hidden states. In contrast, execution produces the actual token-level reasoning steps that are visible, verifiable, and scored by the reward model, which naturally correspond to observed states in the HMM formulation. This separation allows the search process, pruning strategy, and reward-model supervision to be applied at both the latent (planning) level and the observable (execution) level, enabling more precise control over reasoning behavior and improving both efficiency and accuracy.
>
> [1] He et. al, 2024: Make LLMs better zero-shot reasoners: Structure-orientated autonomous reasoning

---

> ### Author Response · Authors · 2025-11-22
> **Response to Reviewer B7cR (2/5)**
>
> (3) How the paradigm could be generalized to other tasks, seem to be unclear.
>
> The plan–execution pattern is not specific to math reasoning or code generation tasks. Many widely studied LLM tasks naturally exhibit a plan–execution structure. For example, multi-hop QA requires first determining which intermediate entity serves as the bridge between the query and answer (plan) before retrieving the relevant facts and synthesizing the answer (execution); tool-augmented reasoning requires deciding which tool to invoke and what input to provide (plan) before calling the tool and interpreting its result (execution); commonsense reasoning often involves inferring the causal or logical structure of the reasoning path (plan) before filling in the concrete reasoning steps based on that structure (execution).
>
> Here are some real examples for the mentioned tasks:
>
> **i. Multi-hop QA Task**
>
> **Question:** In what year was the creator of the current arrangement of the ‘Simpson’s Theme’ born?
>
> **Step 1: Plan:** Identify who created the current arrangement of the Simpson’s Theme.
>
> **Execution:** The current arrangement of the Simpson’s Theme was created by Alf Clausen.
>
> **Step 2: Plan:** Find the birth year of this person.
>
> **Execution:** Alf Clausen was born in 1941. So the answer is 1941.
>
> **ii. Commonsense QA Task**
>
> **Question:** To locate a choker not located in a jewelry box or boutique where would you go?
> A: jewelry store; B: neck; C: jewelry box; D: jewelry box; E: boutique
>
> **Step 1: Plan:** Understand what a “choker” is and where it normally is when being worn. Execution: A choker is a type of necklace that is typically worn around a person’s neck.
>
> **Step 2: Plan:** Use the condition “not located in a jewelry box or boutique” to eliminate options and find where it would be instead.
>
> **Execution:** If the choker is not in a jewelry box or boutique, the natural remaining place to find it is on someone’s neck, so the correct choice is B: neck.
>
> **iii. Tool/Function calling tasks**
>
> **Query:** Find the current temperature in the capital city of France and determine whether it is higher than 10°C.
>
> **Step 1: Plan:** Identify the capital city of France using a knowledge lookup tool. This information is needed before querying the weather.
>
> **Execution:** Call CountryInfoAPI(country='France') → returns capital = "Paris".
>
> **Step 2: Plan:** Using the capital city obtained in Step 1, query a weather API to get the current temperature in Paris, then compare it with 10°C.
>
> **Execution:** Call WeatherAPI(city='Paris', date='today') → return temperature = 12°C. Since 12°C > 10°C, the answer is yes, it is higher than 10°C.
>
> These examples demonstrate that the plan–execution structure naturally emerges in a wide variety of LLM tasks beyond the domains we evaluate in the main paper.
>
> To apply our paradigm to a new reasoning task, one can follow a workflow similar to what we have done in math and code tasks:
>
> 1. Identify or formulate an appropriate plan/execution format that captures the high-level intent (planning) and low-level realization (execution) for the task.
>
> 2. Roll out model trajectories under this format to obtain step-level data that can be used to supervise reward modeling.
>
> 3. Train a reward model (or reuse an existing one) to score planning steps and execution steps based on this data.
>
> 4. Perform dual-phase search during inference, where planning explores diverse high-level options and execution expands them into concrete solutions under reward guidance.
>
> This workflow requires only adapting the representation of planning and execution for the task, while the underlying search algorithm and training procedure remain unchanged.

---

> ### Author Response · Authors · 2025-11-22
> **Response to Reviewer B7cR (3/5)**
>
> > Weakness 2. Need ablation studies to verify the effectiveness of each component of the method.
>
> **Response:** Thanks for your suggestions. In the following, we summarize our existing ablation study results and also provide some additional experiments.
>
> Ablation studies which are already considered:
>
> 1. **With vs. without separating into planning/execution during sampling and scoring.** We have presented and compared the results of dual phase search and single phase search (when there is no adaptive budget allocation) and labeled them as DREAM and Beam Search in Figure 4 and 5. The results show that separating planning and execution consistently improves performance given the same level of total budget.
>
> 2. **With vs. without adaptive budget allocation in Dual Phase Search.** We have presented the results of using / not using adaptive budget allocation in Figure 4 and 5, which are labelled as DREAM+ and DREAM respectively. Across most settings, adaptive budgeting leads to a more favorable accuracy–efficiency trade-off, confirming its effectiveness.
>
> 3. **Synergistic effect between dual-phase search and dynamic budget allocation.** In Appendix C.2, we present results comparing DREAM and DREAM(+) with standard Beam Search and Beam Search equipped with adaptive budget allocation. The results show that adaptive budgeting improves both types of methods, but the gains are consistently larger when combined with dual-phase search. This illustrates that dual-phase search and adaptive budgeting play complementary roles and reinforce each other when used together.
>
> In summary, these ablation studies show that the **main performance gain** of our approach comes from the proposed **dual-phase search with dual-phase reward**, while the adaptive budget allocation brings additional improvements. Moreover, adaptive budgeting complements the dual-phase search, and their combination yields the most favorable accuracy–efficiency trade-off.
>
> To provide a more comprehensive ablation studies, we further provide additional investigation which are presented below:
>
> (1). Only Plan Reward / Only Exec Reward vs Plan + Exec Reward
>
> In Table 12 below, we report the performance when only the planning phase or only the execution phase is sampled with multiple candidates and scored by the reward model, compared with our full dual-phase sampling strategy. The results demonstrate that, under the same level of computation budget, scaling both planning and execution yields better performance than scaling only one of the two phases. This highlights the importance of scaling both phases jointly rather than focusing on scaling either one alone.
>
> Table 12  Only Plan Reward / Only Exec Reward vs Plan + Exec Reward (GSM8K, Qwen2.5-MATH-1.5B )
>
> | plan and execution |       | only plan |       | only exe |       |
> |--------------------|-------|-----------|-------|----------|-------|
> | acc                | tokens| acc       | tokens| acc      | tokens|
> | 85.29%             | 333.40| 79.76%    | 272.60| 82.64%   | 307.08 |
> | 90.98%             | 679.30| 86.81%    | 537.59| 85.90%   | 546.72 |
> | 92.11%             | 1304.99| 90.83%   | 1115.38| 90.37%  | 1138.29 |
> | 93.56%             | 2573.65| 90.45%   | 2808.89| 91.74%  | 2205.39 |
>
> (2). Following the comments of Reviewer 7wYR, we also provide ablation studies to (i) Justify our choice for $\tau_{e_1}$=$\tau_{p_1}$ and (ii) Analyze how different values of  $\tau_{e_1}$/$\tau_{p_1}$ and  $\tau_{e_2}$/$\tau_{p_2}$ influence the performance of DREAM+.
>
> (i) In Table 13, we demonstrate the performance of DREAM+ when $\tau_{e_1}$ and $\tau_{p_1}$ take different values or the same value. The results indicate that using the same threshold ($\tau_{p_1}$=$\tau_{e_1}$) yields slightly better performance. This is expected because different thresholds can lead to imbalanced pruning across the two phases. Since both planning and execution are essential for producing a good reasoning step, pruning one phase more aggressively may either discard valid plans too early or prevent promising plans from being fully explored. Using the same threshold keeps the pruning strength aligned across phases and avoids such imbalance.
>
> Table 13: Performance of DREAM in gsm8k with Qwen2.5-MATH-1.5B-Instruct when $\tau$s take different values. (We set $N_1=N_2=4$, $n_1=n_2=2$)
>
> | **$\tau_2$ = 0.7** |       |        | **$\tau_2$ = 0.9** |       |        |
> |--------------|-------|--------|--------------|-------|--------|
> | *$\tau_{p_1}$/ $\tau_{e_1}$* | acc   | tokens | *$\tau_{p_1}$/ $\tau_{e_1}$* | acc   | tokens |
> | 0.9/0.8      | 89.40% | 501.276 | 0.99/0.9     | 90.00% | 569.846 |
> | 0.8/0.9      | 89.00% | 492.466 | 0.9/0.99     | 91.20% | 525.868 |
> | 0.85/0.85    | 90.00% | 492.182 | 0.95/0.95    | 91.60% | 522.408 |
>
> (will be continue in the next response...)

---

> ### Author Response · Authors · 2025-11-22
> **Response to Reviewer B7cR (4/5)**
>
> Continue from response (3/5)
>
> (ii)  We present additional analysis on how different choices of the thresholds $\tau_{e_1}, \tau_{p_1}, \tau_{e_2}, \tau_{p_2}$ affect the performance of DREAM+. The results are shown in Table 13. The left block reports results under a fixed high early stop threshold ($\tau_{e_1}=\tau_{p_1}=0.99$) while varying the extra-budget thresholds $\tau_{e2}/\tau_{p2}$. The right block reports results under a fixed low extra-budget thresholds ($\tau_{e_2}=\tau_{p_2}=0.3$) while varying the early stop thresholds $\tau_{e_1}/\tau_{p_1}$.
>
> From the results in Table 13, we observe two consistent trends.
>
> First, when $\tau_1$ is fixed, decreasing $\tau_2$ leads to lower accuracy and
> fewer generated tokens. This result is expected, as a higher $\tau_2$ imposes a stricter extra budget standard, causing the model to more frequently determine that the current candidates are not sufficient and therefore allocate more budget to explore further. As $\tau_2$ decreases, fewer steps trigger budget expansion, reducing exploration and slightly lowering accuracy.
>
> Second, when $\tau_2$ is fixed, decreasing $\tau_1$ also reduces accuracy and the number of generated tokens. This is because the thresholds $\tau_1$ determine whether the remaining budget for a step should continue to be used. A higher $\tau_1$ corresponds to a stricter stopping criterion and encourages generating more candidates within the step, thereby resulting in higher accuracy and higher computation costs.
>
> Overall, $\tau_1$ and $\tau_2$ control different aspects of intra-step search, and their effects on accuracy and efficiency follow intuitive patterns.
>
> Table 13: Performance of DREAM+ with different values of thresholds (GSM8k, Qwen2.5-MATH-1.5B-Instruct)
>
> | **$\tau_{e_1}$ = $\tau_{p_1}$ = 0.99** ||| **$\tau_{e_2}$ = $\tau_{p_2}$=0.3** |||
> |-|-|-|--|-|-|
> | $\tau_{e_2}$ / $\tau_{p_2}$| acc   | tokens | $\tau_{e_1}$/$\tau_{e_1}$| acc  | tokens |
> | 0.9/0.9| 0.92  | 538.754| 0.99/0.99| 0.892 | 489.742|
> | 0.7/0.7| 0.902 | 520.430 | 0.8/0.8| 0.884 | 472.432|
> | 0.5/0.5| 0.896 | 506.496| 0.6/0.6| 0.878 | 467.584|
> | 0.3/0.3| 0.892 | 489.742| 0.4/0.4| 0.876 | 465.156|
>
> > Weakness 3: There are too many heuristic designs, such as the width and budgets. All these factors make the proposed method hard to generalize.
>
> **Response:** We thank the reviewer for highlighting the concern on heuristic designs and potential generalization issues. While DREAM introduces several hyperparameters, we would like to clarify that these parameters serve different purposes and are not all task-specific heuristics.
> Specifically, they fall into two categories:
>
> (1) Budget-related parameters (including the beam with $n_1$, $n_2$ and sample size $N_1$, $N_2$) These parameters are not hyperparameters that require tuning, but simply user-specified controls of how much inference budget to allocate per example.
>
> (2) Threshold-related parameters (e.g., early-stop thresholds $\tau_{e_1}$, $\tau_{p_1}$, and reward-based budget assignment criteria $\tau_{e_2}$, $\tau_{p_2}$) These parameters are more task-sensitive, but we found that their optimal ranges are model-agnostic. In our main experiments in Section 4.2, we use a small calibration set with LLaMA to determine the suitable values for both datasets, and then apply the same thresholds directly to the experiments with other models (e.g., Qwen-MATH and DeepSeek-Math and) without any further tuning, while achieving stable performance across datasets.
>
> Therefore, although DREAM exposes several hyperparameters, they either serve as compute controls or require only once-per-dataset calibration, and they generalize well across models.
>
> > Question 1: How the model behaves when not explicitly decomposing the search into planning and execution (paradigm Figure 3 (c) (**should actually be (b) here**))?
>
> **Response:** The behavior of the model without explicitly decomposing the search into planning and execution corresponds to the paradigm in Figure 3(b), which is denoted as Beam Search in our experiments. As reported in Section 4.2.1, the three paradigms in Figure 3(b)(c)(d) are evaluated and reported as Beam Search, DREAM and DREAM+, respectively. According to the results in Figure 4, Beam Search’s performance is consistently worse than DREAM and DREAM+, demonstrating the benefit of explicit dual-phase decomposition.

---

> ### Author Response · Authors · 2025-11-22
> **Response to Reviewer B7cR (5/5)**
>
> > Question 2: How do you instruct the model to do planning and execution? Have you trained the model to do it?
>
> **Response:** As we describe in Section 3.1, the model is not trained to perform planning and execution. Instead, we use few-shot prompting to guide the base LLM to generate reasoning in the plan–execution format. This prompting strategy is sufficient for the model to follow the desired structure without any additional fine-tuning. We have updated the revised manuscript to include the exact prompts in Appendix D.
> We also emphasize that, unlike other test-time scaling approaches that require fine-tuning or additional training of LLMs,our method does not require any LLM fine-tuning, making it lightweight and easily applicable across different base models.

---

> > ### Comment · Reviewer_B7cR · 2025-11-25
> >
> > Thank you for the detailed response and additional experimental results, really appreciate the ablation studies.
> >
> > After reading the response, I think the vague definition of planning and execution remains. And authors can only explain these concepts through examples.
> >
> > The basic assumption in this work, more formally speaking, should be **LLM reasoning mixes different patterns**. On top of that, the claim in this work should be **disentangling these patterns and applying separate guidance are is better than applying a mixed guidance**. Here are the questions to think about:
> > - How are different reasoning patterns mixed? Hierarchical? Sequential? Or parallel? Make this assumption clear and shape your method in the corresponding way, rather than just say *dual*.
> > - How to disentangle these patterns? In this work, they are  human defined and separated via prompting.
> > - Why disentangling and separate guidance are beneficial? The motivation is not so strong in this work, and authors just claim that planning is good and we should do that.
> > - How to obtain the guidance for each reasoning pattern? Partially answered in this work
> > - How to apply the guidance separately? Main focus in this work.
> >
> > It can be seen that this paper directly jumps to the last question without really thinking through the first few. I think the current version still needs lots of improvement and I decide to maintain my rejection assessment. But hope these suggestions could help the authors to improve the work.

---

> ### Author Response · Authors · 2025-11-27
> **Reply to Reviewer B7cR (1/2)**
>
> We sincerely appreciate the reviewer’s feedback. In response to your followup questions, we provide the following clarifications and **have revised our manuscript accordingly** to explicitly incorporate these points.
>
> (1) **Structure of the reasoning pattern.**
>
> Following recent latent-variable formulations of reasoning (e.g., He et al., 2024, Tutunov et al., 2023, Xie et al., 2021), we view reasoning as **a hierarchical process** which interleaves two structured reasoning roles: (i) global structural decisions (planning) and (ii) local step derivation (execution). This is similar to the hierarchical structure with the hidden/observed states in a Hidden Markov Model. Our “planning” phase represents the latent structural pattern, and the “execution” phase represents the observable realization of that structure.
> **We provide a formal definition grounded in their functional roles in generation, which can be found in Section 3.1 in our revised manuscript.**
>
> (2) **Why do we use prompting to disentagle this pattern?**
>
> Although reasoning inherently includes structural (planning) and derivational (execution) components as we defined above, current LLMs do not explicitly separate them and these behaviors are often expressed in a mixed and entangled manner within regular generation. Therefore, in our work, human-designed prompting serves as a natural and effective mechanism to elicit and disentangle these reasoning roles. Prompts do not artificially inject new reasoning behavior. Instead, they leverage the LLM’s instruction following capability to help reveal and structure the reasoning.
>
> (3) **Why is disentangling and separate guidance beneficial?**
>
> To provide a more convincing justification for the separate guidance, in the following, we provide some empirical results demonstrating that **planning and execution exhibit fundamentally different behaviors and therefore different error patterns**. In short, planning steps are subjected to higher uncertainty, but their errors are usually less fatal. On the other hand, execution is simpler, but the calculation error in the math problem can be carried over in the later steps, being more harmful to the reasoning process.
> To justify the above, we provide the following two experiment results. First, we use the perplexity as a metric to evaluate the confidence of the LLM towards the planning and the execution step. The results are summarized in Table 1.
>
> Table 1
> ||| mean   | median | std| min| max|
> |-|--|-|-|-|-|-|
> | **perplexity**  | plan | 5.4646 | 3.8493 | 6.456  | 1.0679 | 88.4054 |
> | | exec |1.2614 | 1.2297 | 0.1831 | 1.0024 | 2.3179  |
>
> From the above table, it is clear that the perplexity of the planning and execution are different, demonstrating that the LLM is in general less confident in the planning step.
>
> Second, we also check the reward value of the planning and execution steps. The results are in Table 2. Based on the numbers, the reward for the execution steps are indeed lower than the planning steps. Our conjecture is that, if the planning makes a mistake, the later planning steps can still search and possibly reach the correct result. However, for the execution steps, once a step gives a wrong number, it is very likely that the later steps will use this number for further calculation. Empirical support for this conjecture is provided in Table 3, where we report the reward distributions of planning and execution across correct and incorrect solutions. The execution rewards show a notably larger discrepancy between correct and incorrect solutions, showing that execution errors are more harmful and more likely to affect the final result.
>
> Table 2
> | | | mean  | median | std| min| max   |
> |-|-|-|-|-|-|-|
> | **reward**  | plan     | 0.804 | 0.957  | 0.274 | 0.008 | 1.000 |
> || exec| 0.768 | 0.957  | 0.319 | 0.002 | 1.000 |
>
> Table 3
> |           |        | mean      | median    | std       | min      | max   |
> |-----------|--------|-----------|-----------|-----------|----------|-------|
> | correct   | plan   | 0.889  | 0.992  | 0.233  | 0.053 | 1     |
> |           | exec  | 0.873     | 0.992     | 0.259     | 0.009    | 1 |
> | incorrect | plan   | 0.735  | 0.945  | 0.342  | 0.008 | 1     |
> |           | exec   | 0.669     | 0.941     | 0.393     | 0.008    | 1 |
>
> Given the above two differences on the planning and execution steps, we hypothesize that separating them during the inference time and dynamically assigning a computation budget could enhance the efficiency of test-time scaling. In particular, **searching over planning steps enables the model to quickly identify promising high-level directions, thereby improving the efficiency of exploration. Meanwhile, searching over execution steps helps the model avoid committing to incorrect derivations and reducing error propagation.**

---

> ### Author Response · Authors · 2025-11-27
> **Reply to Reviewer B7cR (2/2)**
>
> (4) **The way we provide guidance for each reasoning pattern**
>
> To obtain guidance for each reasoning pattern, we train separate reward models that correspond directly to the two roles defined above. Specifically, $PRM_{plan}$ provides feedback for structural decisions $z_i$, and $PRM_{exec}$ provides feedback for token-level derivations $x_i$. Both reward models are trained using supervised signals derived from partial-trajectory rollouts, but each model focuses on a different aspect of reasoning quality: $PRM_{plan}$ evaluates whether the selected subgoal or strategy is promising, while $PRM_{exec}$ evaluates the correctness of the local derivation.
> This separation aligns naturally with the hierarchical definitions of planning and execution, and allows each phase to be guided by the supervision most relevant to its functional role.
>
> References:
>
> Pengfei He, Zitao Li, Yue Xing, Yaling Li, Jiliang Tang, Bolin Ding. Make LLMs better zero-shot reasoners: Structure-orientated autonomous reasoning. arXiv preprint arXiv:2410.19000 (2024).
>
> Rasul Tutunov, Antoine Grosnit, Juliusz Ziomek, Jun Wang, and Haitham Bou-Ammar. Why can large language models generate correct chain-of-thoughts? arXiv preprint arXiv:2310.13571, 2023.
>
> Sang Michael Xie, Aditi Raghunathan, Percy Liang, and Tengyu Ma. An explanation of in-context learning as implicit bayesian inference. arXiv preprint arXiv:2111.02080, 2021.

---

### Official Review · Reviewer_S4PF · 2025-10-29

**Soundness:** 2
**Presentation:** 3
**Contribution:** 2
**Rating:** 4
**Confidence:** 3

**Summary:**

This manuscript proposes the DREAM method, which guides large language models to decouple planning and execution steps during reasoning, and to select each step using its corresponding reward model. In addition, a dynamic sampling method is designed to improve the efficiency of reasoning.

**Strengths:**

1. By explicitly separating the planning and execution steps, DREAM is expected to further improve sampling efficiency with the help of the PRM.

2. The proposed adaptive sampling budget allocation strategy is reasonable.

**Weaknesses:**

1. The proposed method has limited application scenarios: on one hand, it requires multiple specialized reward models to provide rewards for plan and execution respectively. This adds extra burden to both training (especially considering the need to construct specific datasets) and inference deployment, making practical application difficult. On the other hand, it can only be applied to tasks that can be formatted into plan and execution stages.

2. The proposed idea is not very innovative — essentially, it just breaks down the original steps into finer ones. More simply, even without formatting outputs into plan and execution, one could instruct the model to generate smaller reasoning steps to achieve more controllable, stage-guided generation.

**Questions:**

What I’m more concerned about is: where exactly is the boundary for step decomposition? Or can the steps be continuously subdivided into even smaller units? What additional problems would finer decomposition bring? And how should we account for the computational cost of the reward model?

---

> ### Author Response · Authors · 2025-11-22
> **Response to Reviewer S4PF (1/4)**
>
> We appreciate your efforts in reviewing our paper and recognizing our contribution to the solid theories. We will revise our paper based on your suggestions, and below are some answers to your comments and questions:
>
> > Weakness 1.1 & Question 4: Using double reward model would add extra burden to both training (especially considering the need to construct specific datasets) and inference deployment. Need to consider the computational cost of the reward model.
>
> **Response:** We acknowledge the reviewer’s concern regarding the use of two reward models. We would like to clarify that this design does not introduce a significant burden in either training or inference.
>
> **For training**: While it is true that extra burden is required to obtain the training data of the reward model, we note that the reward model only needs to be trained once and can then be reused across different datasets and backbone LLMs. As shown in Sections 4.2 and 4.3, the same reward model generalizes well across models and datasets, so no dataset-specific retraining is required. Moreover, in practical deployments, the training cost can be further reduced: the plan-reward and execution-reward data can be combined to train a single unified reward model, and our experiments show that this universal model performs comparably to specialized models (see the table below).
>
> Table 8. Comparison of using a specialized reward model or a univesal reward model in DREAM+ (GSM8K, Qwen2.5-MATH-1.5B-Instruct)
>
> | Specialized model |        | Universal model |        |
> |-------------------|--------|-----------------|--------|
> | acc               | tokens | acc             | tokens |
> | 0.8529        | 333.39 | 0.8503        | 328.73 |
> | 0.9098          | 679.29 | 0.9067       | 638.13 |
> | 0.9211            | 1304.99 | 0.9204         | 1280.07 |
> | 0.9356        | 2573.65 | 0.9325        | 2568.99 |
>
> **For inference:** To provide a concrete comparison of inference-time computation costs, we report the FLOPs of LLM generation (FLOP-gen) and reward-model evaluation (FLOP-rew) for different methods under representative configurations. The table below presents the results for Llama3-8B-Instruct on GSM8K, with all methods evaluated using a tree-search width of 16. Due to the limited rebuttal time, we include only a subset of the FLOPs analysis here. We will update the manuscript to include the full computation-cost results for all models and datasets in the revised version.
>
> In Table 9, we further include an additional configuration of DREAM using a smaller search width (8). This setting approximately matches the compute budget of the baseline methods, while the width-16 setting matches their prediction accuracy.  From the table, we observe that FLOPs for LLM generation (FLOP-gen) increase approximately linearly with the number of generated tokens, which aligns with the nature of tree-based test-time scaling: broader search leads to more candidate generations, and thus more decoding steps. For the reward-model FLOPs (FLOP-rew), DREAM (+) requires roughly twice the reward computations compared with methods that do not separately score planning and execution. However, even after this doubling, the total FLOP-reward remains less than 1/10 of FLOP-gen. This indicates that dual-phase scoring adds only a small fraction of overall compute.
>
> Table 9: Computation of each method in GSM8k using LLaMA3-8B-Instruct
>
> |  | DREAM   | Beam Search | REBASE  | Majority Volte     | DREAM+(1)   | DREAM+(2)  |
> |----------|---------|-------------|---------|---------|----------|---|
> | acc      | 94.62%  | 93.53%      | 93.86%  | 87.34%  | 95.38%| **93.86%** |
> | tokens   | 2442.21 | 2339.38     | 2426.49 | 2598.10 | 2611.44| **1508.04** |
> | FLOPS-gen| 4.34E+15| 4.17E+15     | 4.31E+15| 4.63E+15| 4.64E+15 |**2.68E+15** |
> | FLOPS-rew| 3.43E+14| 1.77E+14     | 1.84E+14| 0       |3.67E+14 |**1.14E+14** |
> | FLOPS-sum| 4.68E+15| 4.35E+15     | 4.49E+15| 4.63E+15| 5.00E+15 |**2.79E+15** |

---

> > ### Author Response · Authors · 2025-11-22
> > **Response to Reviewer S4PF (2/4)**
> >
> > > Weakness 1.2: Only apply to task which can be divided into plan/execution
> >
> > Response: Thank you for raising the concern regarding the applicability of our method. We agree that our method is designed for tasks where the reasoning process can be naturally expressed through a planning step followed by an execution step. However, we would like to highlight that this structure is not limited to a narrow set of tasks,  because in real practice, many reasoning problems inherently follow a plan–execution pattern. Beyond the math and code-generation tasks discussed in our paper, we provide additional examples below demonstrating how other common tasks can also be decomposed into planning and execution.
> >
> > **(1). Multi-hop QA Task**
> >
> > **Question:** In what year was the creator of the current arrangement of the ‘Simpson’s Theme’ born?
> >
> > **Step 1: Plan:** Identify who created the current arrangement of the Simpson’s Theme.
> >
> > **Execution:** The current arrangement of the Simpson’s Theme was created by Alf Clausen.
> >
> > **Step 2: Plan:** Find the birth year of this person.
> >
> > **Execution:** Alf Clausen was born in 1941. So the answer is 1941.
> >
> > **(2). Commonsense QA Task**
> >
> > **Question:** To locate a choker not located in a jewelry box or boutique where would you go?
> >
> > A: jewelry store; B: neck; C: jewelry box; D: jewelry box; E: boutique
> >
> > **Step 1: Plan:** Understand what a “choker” is and where it normally is when being worn.
> >
> > **Execution:** A choker is a type of necklace that is typically worn around a person’s neck.
> >
> > **Step 2: Plan:** Use the condition “not located in a jewelry box or boutique” to eliminate options and find where it would be instead.
> >
> > **Execution:** If the choker is not in a jewelry box or boutique, the natural remaining place to find it is on someone’s neck, so the correct choice is B: neck.
> >
> > **(3). Tool/Function calling tasks**
> >
> > **Query:** Find the current temperature in the capital city of France and determine whether it is higher than 10°C.
> >
> > **Step 1: Plan:** Identify the capital city of France using a knowledge lookup tool. This information is needed before querying the weather.
> >
> > **Execution:** Call CountryInfoAPI(country='France') → returns capital = "Paris".
> >
> > **Step 2: Plan:** Using the capital city obtained in Step 1, query a weather API to get the current temperature in Paris, then compare it with 10°C.
> >
> > **Execution:** Call WeatherAPI(city='Paris', date='today') → return temperature = 12°C. Since 12°C > 10°C, the answer is yes, it is higher than 10°C.
> >
> > These examples demonstrate that the plan–execution structure naturally emerges in a wide variety of LLM tasks beyond the domains we evaluate in the main paper.

---

> ### Author Response · Authors · 2025-11-22
> **Response to Reviewer S4PF (3/4)**
>
> > Weakness 2, Question 1-3: Novelty about the proposed decomposition, and where exactly is the boundary for step decomposition?
>
> **Response:** Thank you for raising these important questions regarding the novelty of our method and the distinction between plan–execution decomposition and simply increasing step granularity. In response, we would like to highlight that plan–execution decomposition is fundamentally different from finer-grained step decomposition. Specifically, increasing granularity means dividing a single reasoning step into smaller sequential segments (e.g., splitting one step into two micro-steps).
>
> In contrast, plan–execution decomposition separates two different types of reasoning behaviors. The planning component focuses on forming high-level strategic decisions or subgoals, while the execution component focuses on concretely carrying out these decisions. Consequently, decomposing a step into plan and execution changes the structure of the reasoning process rather than merely adjusting the level of granularity.
>
> To better illustrate the difference, we provide real examples demonstrating both strategies and the combination of both strategies:
>
> Q: Jack deposited $4000 in a bank with a 1% annual interest rate. What will be the total amount after one year?
>
> **1. Standard CoT:**
>
> Step 1: With 1% interest rate, after the first year, the total value is 10000×(1+1%)=10100
>
> Step 2: With 2% interest rate, after the second year, the total value is 10100(1+2%)=10302
>
> **2. CoT with increased granularity:**
>
> Step 1: With 1% interest rate, the first year’s interest is 10000*1%=100.
>
> Step 2: The total value after the first year is 10000+100=10100.
>
> Step 3: With 2% interest rate, the second year’s interest 10100*2% = 202.
>
> Step 4: The total value after the second year is 10100+202=10302.
>
> **3. CoT with plan/execution decomposition:**
>
> Step 1: Determine the total value after the first year:
>
> \- Given that the interest rate of the 1st year is 1%. The total value becomes 10000×(1+1%)=10100
>
> Step 2: Determine the total value after the second year:
>
> \- Given that the interest rate of the 1st year is 2%. The total value becomes 10100×(1+2%)=10302.
>
> **4. CoT with plan/execution decomposition + increased granularity:**
>
> Step 1: Determine the first year’s interest:
>
> \- Given that the interest rate of the 1st year is 1%. The first year’s interest is 1% x 10000=100
>
> Step 2: Determine the total value after the first year:
>
> \- After the first year, the interest is 10000+100=10100
>
> Step 3: Determine the second year’s interest:
>
> \- Given that the interest rate of the 2nd year is 2%. The second year’s interest is 2% x 10100=202
>
> Step 4: Determine the total value after the second year:
>
> \- After the second year, the total value becomes 10100+202 = 10302.
>
> To better demonstrate that the improvement of DREAM comes from structural plan–execution decomposition rather than simply splitting steps into smaller units, we compare all the four CoT variants shown in the above example using DREAM under different beam widths. The numerical results are provided in Table 10 and visualized in Figure 7 (shown in modified pdf).
>
> Table 10. Comparision of four CoT format. (on MATH dataset with LLaMA3.1-8B-Instruct)
>
> | width | standard || granularity || plan / execution | | plan/execution+granularity |        |
> |-|-|-|-|-|-|-|-|-|
> || acc| tokens | acc| tokens | acc| tokens | acc | tokens |
> | 2     | 0.48     | 811.938 | 0.484| 1011.53 | 0.522           | 949.49 | 0.538| 1302.44 |
> |4| 0.568| 1578.09 | 0.57| 1902.53 | 0.61| 2021.16 | 0.614| 2631.79 |
> |8| 0.61 | 3170.16 | 0.638| 3805.05  | 0.668| 3854.31 | 0.65| 5149.69 |
> |16| 0.658    | 6188.42 | 0.694| 7872.97 | 0.708| 7776.57 | 0.714| 10587.93 |
>
> Based on the results, we have the following observations.
>
> First, when comparing entries within each row of the table i.e., under the same searching width, we observe both increasing granularity and applying plan–execution decomposition can improve accuracy, but both approaches also require more tokens and therefore higher computation. However, the accuracy–efficiency trade-off shown in Figure 1 reveals a clear difference: there is no obvious gain from finer-grained steps in terms of accuracy-efficiency trade-off, whereas the gain from plan–execution decomposition is obvious.
>
> (will be continued in next response...)

---

> ### Author Response · Authors · 2025-11-22
> **Response to Reviewer S4PF (4/4)**
>
> (continue from response 3/4)
>
> Second, when the beam width is fixed, combining increased granularity with plan–execution decomposition leads to higher accuracy than using plan–execution alone. However, this combined strategy substantially increases the number of generated tokens and makes the reasoning trace significantly more verbose. As a consequence, the computational cost grows faster than the improvement in accuracy. When examining the accuracy–efficiency trade-off, we do not observe advantages beyond using plan–execution decomposition alone.
>
> In summary, the results confirm that the benefit of DREAM does not come from simply making the reasoning steps more fine-grained, but from the structural decomposition that separates high-level subgoal planning from low-level execution, enabling more effective exploration and pruning and ultimately yielding a better accuracy–efficiency trade-off.
>
> **Additional justification about the novelty:** We would like to highlight that, in addition to introducing the plan–execution decomposition, another key innovation of our method is the **adaptive budget allocation mechanism**, which dynamically allocates sampling efforts based on the estimated promise of intermediate candidates. As evidenced in our experiments, this adaptive allocation leads to more efficient use of computation and further improves the accuracy–efficiency trade-off beyond what can be achieved by decomposition alone.

---

### Official Review · Reviewer_7wYR · 2025-10-31

**Soundness:** 2
**Presentation:** 2
**Contribution:** 2
**Rating:** 4
**Confidence:** 4

**Summary:**

The paper introduces **DREAM**, a framework that explicitly decomposes reasoning into two phases—planning and execution. Separate reward models are trained for planning and for execution to conduct independent tree-structured search and pruning. The authors further propose a dual-threshold adaptive compute budget: terminate early when any candidate surpasses an upper threshold to save computation, and trigger additional sampling when all candidates fall below a lower threshold, thereby reallocating compute from easy steps to harder ones and improving the accuracy–efficiency trade-off. On mathematical reasoning and code generation tasks, DREAM demonstrates strong cross-model and cross-dataset generalization.

**Strengths:**

The paper clearly delineates two major shortcomings of current tree-search-based test-time scaling (TTS): (i) planning and execution are evaluated jointly, and (ii) the step budget is fixed. It proposes DREAM, which decouples reasoning into planning and execution phases, equips each phase with a dedicated reward model, and applies dynamic step-level budgeting, thereby reducing ineffective exploration and focusing computation on critical steps.

**Weaknesses:**

1. Lacks theoretical and empirical analysis of the thresholds (e.g., $\tau_{p1}$ and $\tau_{p2}$), which are likely to strongly affect resource allocation and accuracy. Paper does not analyze their impact.

2. Experimental details are insufficient. The settings for $\tau_{p1}$, $\tau_{p2}$, $m_1$, and $m_2$ are missing, making the experiments difficult to reproduce.

3. Missing evaluations on more challenging mathematical reasoning datasets and stronger reasoning models, which makes it hard to assess the method’s generalization.

4. Typo: in Figure 2, the “Plan–Execution Format” contains two instances of “Q2.”

**Questions:**

1. In the main results, REBASE underperforms Beam Search on mathematical datasets (especially MATH), despite achieving better compute allocation. Please explain the reasons for this discrepancy.

2. DREAM verifies each step in two phases. Does this increase test-time compute? Please provide a detailed test-time compute report, including FLOPs for the policy model and the PRM, as well as end-to-end inference latency.

3. Is the setting $\tau_{p1}=\tau_{e1}$ reasonable? The planning and execution phases differ substantially in sampling distribution and inter-sample dependency, raising concern that DREAM may simply split a step and apply finer-grained rewards without truly distinguishing between planning and execution.

---

> ### Author Response · Authors · 2025-11-22
> **Response to Reviewer 7wYR (1/4)**
>
> We appreciate your efforts in reviewing our paper and recognizing our contribution in the solid theories. We will revise our paper based on your suggestions, and below are some answers to your comments and questions:
>
> > Weakness1 & Question 3: Theoretical and empirical analysis on the influence of the thresholds, and why $\tau_{e_1}$=$\tau_{p_1}$.
>
> **Response**: Thanks for your suggestion and question. Below we provide (1). Justification about why $\tau_{e_1}$=$\tau_{p_1}$; and (2). Theoretical and empirical analysis on how different choices of the thresholds $\tau_{e1}, \tau_{p1}, \tau_{e2}, \tau_{p2}$ affect the performance of DREAM+. We have also updated our manuscript to include these analysis in Appendix C.3 and C.4.
>
> **1. Justification about why $\tau_{e_1}$=$\tau_{p_1}$.**
> To justify why $\tau_{e_1}$=$\tau_{p_1}$, empirically, as shown in the following table, using the same threshold ($\tau_{e_1}$=$\tau_{p_1}$) yields slightly better performance.
>
> Table 1: Performance of DREAM in gsm8k with Qwen2.5-MATH-1.5B-Instruct when $\tau$s take different values. (We set $N_1=N_2=4$, $n_1=n_2=2$)
>
> | **$\tau_2$ = 0.7**||| **$\tau_2$ = 0.9**|||
> |-|-|-|-|-|-|
> | *$\tau_{p_1}$/ $\tau_{e_1}$* |acc| tokens | *$\tau_{p_1}$/ $\tau_{e_1}$* | acc| tokens |
> | 0.9/0.8| 89.40% |501.276 |0.99/0.9 | 90.00% | 569.846 |
> | 0.8/0.9| 89.00% |492.466 |0.9/0.99| 91.20% | 525.868 |
> | 0.85/0.85| 90.00% |492.182 |0.95/0.95| 91.60% | 522.408 |
>
>
> The intuition behind this observation is twofold.
>
> (1). Although planning and execution operate on different sampling spaces, their reward scores share a comparable numerical scale: both reward models evaluate step-level reasoning quality (whether a subgoal is sensible, or whether a subexecution is correct).  Since the rewards are trained under the same supervision paradigm, applying the same threshold yields consistent pruning behavior in both phases.
>
> (2). Different thresholds can easily create imbalanced pruning: Both planning and execution are essential for producing a good reasoning step. If one phase is pruned more aggressively than the other, the search becomes unbalanced. In such cases, either valid plans are discarded too early, or promising plans cannot be properly expanded. Therefore, using the same threshold keeps the pruning strength aligned across phases and prevents either side from dominating the search.
>
> **2. Theoretical and empirical analysis on the influence of the thresholds**
>
> **(1) Empirical Results.** In Table 2, we provide empirical results on how different values of $\tau_{e1}, \tau_{p1}, \tau_{e2}, \tau_{p2}$ influence the performance of DREAM+.
>
> Table 2: Performance of DREAM+ with different values of thresholds (GSM8k, Qwen2.5-MATH-1.5B-Instruct)
>
> | **$\tau_{e_1}$ = $\tau_{p_1}$ = 0.99** ||| **$\tau_{e_2}$ = $\tau_{p_2}$=0.3** |||
> |-|-|-|--|-|-|
> | $\tau_{e_2}$ / $\tau_{p_2}$| acc   | tokens | $\tau_{e_1}$/$\tau_{e_1}$| acc   | tokens |
> | 0.9/0.9| 0.92  | 538.754| 0.99/0.99| 0.892 | 489.742|
> | 0.7/0.7| 0.902 | 520.430 | 0.8/0.8| 0.884 | 472.432|
> | 0.5/0.5| 0.896 | 506.496| 0.6/0.6| 0.878 | 467.584|
> | 0.3/0.3| 0.892 | 489.742| 0.4/0.4| 0.876 | 465.156|
>
> The left block reports results under a fixed high early step threshold ($\tau_{e1}=\tau_{p1}=0.99$) while varying the extra-budget thresholds $\tau_{e2}/\tau_{p2}$. The right block reports results under a fixed low extra-budget thresholds ($\tau_{e2}=\tau_{p2}=0.3$) while varying the early stop thresholds $\tau_{e1}/\tau_{p1}$.
>
> From the results in Table 2, we observe two consistent trends.
>
> First, when $\tau_1$ is fixed, decreasing $\tau_2$ leads to lower accuracy and  fewer generated tokens. This result is expected, as a higher $\tau_2$ imposes a stricter extra budget standard, causing the model to more frequently determine that the current candidates are not sufficient and therefore allocate more budget to explore further. As $\tau_2$ decreases, fewer steps trigger budget expansion, reducing exploration and slightly lowering accuracy.
>
> Second, when $\tau_2$ is fixed, decreasing $\tau_1$ also reduces accuracy and the number of generated tokens. This is because the thresholds $\tau_1$ determine whether the remaining budget for a step should continue to be used. A higher $\tau_1$ corresponds to a stricter stopping criterion and encourages generating more candidates within the step, thereby resulting in higher accuracy and higher computation costs.
>
> Overall, $\tau_1$ and $\tau_2$ control different aspects of intra-step search, and their effects on accuracy and efficiency follow intuitive patterns.

---

> ### Author Response · Authors · 2025-11-22
> **Response to Reviewer 7wYR (2/4)**
>
> **(2) Theoretical Analysis: Markov-based Analysis of the Threshold Behaviors**
>
> To build a theoretical understanding of how the thresholds $\tau_1$ and $\tau_2$ influence DREAM+, we begin by modeling the within-step search as a **simplified Markov process**.
>
> To begin, we model the single step generation in DREAM+ as a simple Markov process with states $\\{1,2,3,...,n,\*\\}$ where state $*$ denotes the end state. For any states $a$ and $b$, let $P(x_{i+1}=a \mid x_i=b)=P_{b\rightarrow a}$. We adopt the following assumptions for the transition matrix:
>
> a. $P_{b\rightarrow b+1}=\frac{1}{2}$ for any $1\leq b<n$
>
> b. $P_{a\rightarrow *}=\frac{1}{2}$ for any $1\leq a \leq n$
>
> c. $P_{n\rightarrow n}=\frac{1}{2}$
>
>  At step $i$, each beam samples multiple candidates for next states and selects the one with the highest reward value based on the reward model. This process can be modeled as: at state $x_i=b$, we draw several independent samples of
> $x_{i+1}$ under a sampling budget $k$, and retain the candidate with the highest reward. We further assume that the reward assigned by the reward model is positively correlated with the transition likelihood toward the end state \*, $P^{\infty}\_{x\_{i+1}\rightarrow \*}$. Then we have the following theorem:
>
> ***Theorem 1**.  Under the assumptions described above, suppose the maximum number of reasoning steps is $M$ ($M$ is sufficiently large). Then starting from state $b$, the probability of eventually reaching the terminal state \* within the remaining $M-i$ steps is: $P^{(M-i)}\_{b\rightarrow \*}=1-2^{-(M-i-b+1)}$. Furthermore, if at step $i$ we sample $k$ candidates for $x_{i+1}$ and keep the one with the highest reward, then the probability of successfully reaching the end state becomes: $P^\*=1-(\frac{1}{2^{M-i-b+1}})^k$.*
>
> Based on the above threshold, we have the following two propositions on the impact of $\tau_1$ and $\tau_2$ towards $P^*$. For simplicity, each proposition considers $\tau_1$ and $\tau_2$ respectively.
>
> **Proposition 1**. Let $p=1-\frac{1}{2^{(M-i-b)}}$. Assume the sampling budget is $k$ and $\tau_2=0$. Then:
>
> (1) If $\tau_1 \in (0,p]$,  the probability of successfully reaching the end states is: $P^*=1-(\frac{1}{2^{M-i-b+1}})$.
>
> (2) If $\tau_1 \in (p,1]$, $P^*=1-(\frac{1}{2^{M-i-b+1}})^k$
>
> When $k>1$, $P^\*$ is higher when $\tau_1 \in (p,1]$.
>
> **Proposition 2**. Let $p=1-\frac{1}{2^{(M-i-b)}}$. Assume that the original sampling budget is $k$, the maximum additional budget triggered by $\tau_2$ is $k'$, and $\tau_1=1$ is fixed. Then:
>
> (1) If $\tau_2 \in (0,p]$,  the probability of successfully reaching the end states is: $P^*=1-(\frac{1}{2^{M-i-b+1}})^k$.
>
> (2) If $\tau_2 \in (p,1]$, $P^\*=1-(\frac{1}{2^{M-i-b+1}})^{(k+k')}$
>
> Thus, $P^\*$ is higher when $\tau_2 \in (p,1]$.
>
> The two proposition provide evidence that, when all the other conditions are fixed, increasing either $\tau_1$ or $\tau_2$ increase the probability of successfully reaching the end state increase.
> This theoretical analysis explains why larger thresholds generally lead to better performance in DREAM+, as they effectively allocate more sampling budget to promising reasoning paths.
>
> For simplicity, our analysis considers the effect of each threshold independently by fixing one and varying the other. Intuitively, increasing both $\tau_1$ and $\tau_2$ at the same time would compound their effects, which tightening selection and increasing the effective sampling budget, thereby further enhancing the probability of reaching the correct solution.
>
> > Weakness 2: Experiments details about the setting of thresholds and budgets.
>
> **Response**: Thank you for the suggestion. Here we provide the detailed descriptions of the settings of thresholds and budgets in our experiments. We have added this content in Appendix B.2 of our paper.
>
> Settings of threshold and budgets. For GSM8K, we use $\tau_{p_1}=\tau_{e_1}=0.99$, $\tau_{p_2}=\tau_{e_2}=0.9$, and $m_1$=$m_2$=2 for all models. For MATH, we set $\tau_{p_1}=\tau_{e_1}=0.9$, $\tau_{p_2}=\tau_{e_2}=0.5$, and $m_1$=$m_2$=2. To control test-time scaling, we vary the sampling budget $N_1$, $N_2$ and the beam widths $n_1$, $n_2$. The candidate values used in our experiments are shown in the table below; each column represents one configuration, corresponding to a single point in the accuracy–budget curves in the main results in Figure 4. Increasing these values proportionally increases inference-time computation.
>
> Table 3
> |||||||
> |-|-|-|-|-|-|
> | **N1** | 2 | 4 | 8 | 16 | 32 |
> | **N2** | 2 | 4 | 8 | 16 | 32 |
> | **n1** | 1 | 2 | 2 | 4  | 4  |
> | **n2** | 1 | 2 | 2 | 4  | 4  |
>
> Notably, we set $N_1=N_2$ and $n_1=n_2$ to keep the search space balanced between the planning and execution phases. Since both phases play important roles in producing a high-quality reasoning step, using symmetric widths prevents the search from being unintentionally biased toward one phase and also avoids introducing additional hyperparameter tuning.

---

> ### Author Response · Authors · 2025-11-22
> **Response to Reviewer 7wYR (3/4)**
>
> > Weakness 3: Need experiments in more challenging datasets and more capable models.
>
> **Response:** We thank the reviewer for the suggestion to include more challenging datasets and more capable LLMs in our main experiments. To address this, we additionally evaluate **Qwen2.5-MATH-72B-Instruct**, one of the strongest math‐reasoning models, on two harder benchmarks: **AMC23** and **GSM8K-Hard**. In these experiments, we use the same reward model as the one used in the experiments of Section 4.2. The results are shown in Table 4 and 5, and the visualization of the accuracy-efficiency trade-off can be found in Figure 7 in the updated pdf.
> From the results we can observe that, DREAM consistently achieves a better accuracy–efficiency trade-off across both datasets compared with the baselines, demonstrating that the benefits of our method generalize to stronger models and more difficult reasoning tasks.
>
> Table 4. Qwen2.5-MATH-72B-Instruct on AMC23
>
> | DREAM |       | Beam Search |       | REBASE |       | Majority Vote |       |
> |-|-|-|-------|-|-|-|-|
> | acc   | tokens| acc         | tokens| acc    | tokens| acc           | tokens|
> | 0.55  | 901.7 | 0.550        | 1206.625 | 0.525 | 710.425 | 0.525 | 1101.425 |
> | 0.675 | 1660.2| 0.625       | 1943.3   | 0.575 | 1306.475 | 0.575 | 1579.4   |
> | 0.675 | 3960.83| 0.650       | 4142.125 | 0.650  | 3837.675 | 0.625 | 3663.275 |
> | 0.7   | 7128.45| 0.650       | 8314.875 | 0.675 | 5503.650  | 0.675 | 6373.825 |
>
> Table 5. Qwen2.5-MATH-72B-Instruct on GSM8k-Hard
>
> | DREAM || Beam Search || REBASE | | Majority Vote ||
> |-|--|-|--|-|---|---|--|
> | acc  | tokens| acc  | tokens| acc    | tokens| acc | tokens|
> | 0.595| 584.345 | 0.535 | 492.91  | 0.5200  | 347.075 | 0.505 | 768.035 |
> | 0.61 | 1190.035| 0.59  | 891.89  |  0.5150 | 735.22  | 0.545 | 1030.26 |
> | 0.64 | 2146.05 | 0.58  | 2001.45 | 0.530  |1541.035 | 0.555 | 2210.765 |
> | 0.65 | 4436.815| 0.625 | 3863.665| 0.540  | 2911.175 | 0.575 | 4281.115 |
>
> > Weakness 4: Typos.
>
> **Response**: Thanks for pointing out the typo. We have fixed it and updated the pdf.
>
> > Question 1: Why REBASE underperforms Beam Search?
>
> **Response**: REBASE operates by generating multiple candidates at each step and allocates the sampling budget for the next step proportionally to their reward-model scores.  Candidates with higher rewards receive more budget, while candidates with lower rewards receive less. In contrast, Beam Search selects the top-n candidates at each step and assigns each of them the same amount of budget for the next step. This difference makes REBASE inherently more greedy and also more sensitive to noise in reward-model predictions.
>
> Given that the reward model has limited size, its predictions on partial reasoning steps are not perfect. To support this, we evaluated the accuracy of our reward model on Math-500: we first generate partial reasoning paths, obtain labels through rollout in the same way as in reward-model training, and then apply the reward model to classify each step as positive/negative. Using 0.5 as the decision threshold for mapping reward-model scores to binary labels (score > 0.5 interpreted as positive and score ≤ 0.5 as negative), we find that the reward-model accuracy is approximately 88.8%, indicating a non-negligible noise level. Therefore, REBASE’s stronger reliance on fine-grained reward-model scores makes it more vulnerable to imperfect predictions, whereas Beam Search benefits from a more conservative allocation strategy.
>
> Given the presence of noise, the reason why it does not significantly impact the performance of our method is that DREAM(+) mainly relies on the relative ordering of candidates rather than precise reward magnitudes when making planning and execution decisions. As long as the reward model can reliably distinguish promising candidates from weak ones, the search procedure remains stable and is only weakly affected by fine-grained prediction noise.
>
> In addition, we would like to highlight that, despite the presence of noise, our reward model is still relatively strong. To support this, we provide results of experiments using the same REBASE procedures but replacing our reward model with the reward model by [1]. We compare the performance in Table 6 below:
>
> Table 6. Comparison of reward models (REBASE on GSM8k with LLaMA3.1-8B-Instruct )
> | Reward by [1] | | Ours   |        |
> |-|--|--|-|
> | acc  | tokens| acc  | tokens|
> | 0.818         | 311.94 | 0.8582 | 298.21 |
> | 0.859         | 627.81 | 0.9234 | 607.83 |
> | 0.8817        | 1261.10| 0.9303 | 1219.96|
> | 0.8984   | 2532.72| 0.9386 | 2426.49|
>
> From the table, we observe that our reward model demonstrates better performance compared with the reward model from [1]. This demonstrates that, despite the presence of noise, our reward model is comparatively stronger and provides more reliable search guidance than the previous approach.
>
> [1] Wang et. al, 2024: Math-Shepherd: Verify and Reinforce LLMs Step-by-step without Human Annotations

---

> ### Author Response · Authors · 2025-11-22
> **Response to Reviewer 7wYR (4/4)**
>
> > Question 2: Need to report computations cost by each method.
>
> **Response:** Thank you for the suggestion to report the computation cost of each method. In Section 4.2 of the main paper, we already present the accuracy–computation trade-off using the number of generated tokens, which is the dominant component of inference cost in autoregressive models. To provide a more complete view that also accounts for the use of the reward model, we additionally report the approximations of FLOPs of LLM generation (FLOP-gen) and the FLOPs of reward-model evaluation (FLOP-rew) for different methods under representative configurations. The table below shows the results for Llama3-8B-Instruct on GSM8K, with all methods evaluated under a tree-search width of 16. Due to the limited rebuttal time, we include only a subset of the FLOPs analysis here. We will update the manuscript to include the full computation-cost results for all models and datasets in the later version.
>
> To provide a direct comparison, in Table 7, we further include an additional configuration of DREAM+ using a smaller search width (8). This setting approximately matches the prediction accuracy of baseline methods, while the width-16 setting matches their computation budget.  From the table, we observe that FLOPs for LLM generation (FLOP-gen) increase approximately linearly with the number of generated tokens, which aligns with the nature of tree-based test-time scaling: broader search leads to more candidate generations, and thus more FLOPs. For the reward-model FLOPs (FLOP-rew), DREAM (+) requires roughly twice the reward computations compared with methods that do not separately score planning and execution. This is expected, since scoring both the plan and the execution naturally requires calling the reward model twice. However, even after this doubling, the total FLOP-reward remains less than 1/10 of FLOP-gen in the implementation of DREAM. This is because reward scoring only requires a single forward pass over the partial sequence without any autoregressive decoding, whereas generative computation must perform multiple sequential forward passes for token-by-token generation. This indicates that dual-phase scoring does not add substantial computing during inference.
>
> Comparing DREAM with baseline search methods, we observe that DREAM requires slightly more FLOPs but correspondingly achieves higher accuracy. More importantly, DREAM+ with adaptive budget allocation achieves a significantly better accuracy–efficiency trade-off. Under a similar accuracy level as beam search or REBASE, DREAM+ requires substantially fewer total FLOPs, demonstrating that adaptive allocation effectively concentrates computation on promising reasoning trajectories and improves computational efficiency.
>
> In summary, while dual-phase scoring introduces additional computation, the overall cost increase is limited, and DREAM+ consistently obtains stronger accuracy under similar or reduced compute budgets. These findings highlight that structuring search into planning and execution, together with adaptive budget allocation, yields a more favorable efficiency–accuracy trade-off than standard single-phase search.
>
>
> Table 7: Computation of each method in GSM8k using LLaMA3-8B-Instruct
>
> |   | DREAM   | Beam Search | REBASE  | Majority Vote     | DREAM+(1)   | DREAM+(2)  |
> |----------|---------|-------------|---------|---------|----------|---|
> | acc      | 94.62%  | 93.53%      | 93.86%  | 87.34%  | 95.38%| **93.86%** |
> | tokens   | 2442.21 | 2339.38     | 2426.49 | 2598.10 | 2611.44| **1508.04** |
> | FLOPS-gen| 4.34E+15| 4.17E+15     | 4.31E+15| 4.63E+15| 4.64E+15 |**2.68E+15** |
> | FLOPS-rew| 3.43E+14| 1.77E+14     | 1.84E+14| 0       |3.67E+14 |**1.14E+14** |
> | FLOPS-sum| 4.68E+15| 4.35E+15     | 4.49E+15| 4.63E+15| 5.00E+15 |**2.79E+15** |

---

### Note · Authors · 2026-01-04

I have read and agree with the venue's withdrawal policy on behalf of myself and my co-authors.